# Hidden long-range memories of growth and cycle speed correlate cell cycles in lineage trees

Erika E Kuchen[1,2†], Nils B Becker[1,2†], Nina Claudino[1,2], Thomas Höfer[1,2*]

[1]Theoretical Systems Biology, German Cancer Research Center (DKFZ), Heidelberg, Germany; [2]Bioquant Center, University of Heidelberg, Heidelberg, Germany

**Abstract** Cell heterogeneity may be caused by stochastic or deterministic effects. The inheritance of regulators through cell division is a key deterministic force, but identifying inheritance effects in a systematic manner has been challenging. Here, we measure and analyze cell cycles in deep lineage trees of human cancer cells and mouse embryonic stem cells and develop a statistical framework to infer underlying rules of inheritance. The observed long-range intra-generational correlations in cell-cycle duration, up to second cousins, seem paradoxical because ancestral correlations decay rapidly. However, this correlation pattern is naturally explained by the inheritance of both cell size and cell-cycle speed over several generations, provided that cell growth and division are coupled through a minimum-size checkpoint. This model correctly predicts the effects of inhibiting cell growth or cycle progression. In sum, we show how fluctuations of cell cycles across lineage trees help in understanding the coordination of cell growth and division.

## Introduction

Cells of the same type growing in homogeneous conditions often have highly heterogeneous cycle lengths (*Smith and Martin, 1973*). The minimal duration of the cell cycle will be determined by the maximal cellular growth rate in a given condition (*Kafri et al., 2016*). However, many cells, in particular, in multicellular organisms, do not grow at maximum rate, and their cycle length appears to be set by the progression of regulatory machinery through a series of checkpoints (*Novak et al., 2007*). While much is known about the molecular mechanisms of cell-cycle regulation, we have little quantitative understanding of the mechanisms that control duration and variability of the cell cycle.

Recently, extensive live-cell imaging data of cell lineages have become available, characterizing, for example, lymphocyte activation (*Mitchell et al., 2018*; *Duffy et al., 2012*; *Hawkins et al., 2009*), stem cell dynamics (*Filipczyk et al., 2015*), cancer cell proliferation (*Spencer et al., 2013*; *Barr et al., 2017*; *Ryl et al., 2017*), or nematode development (*Du et al., 2015*). Such studies across many cell types have found that cycle lengths are similar in sister cells, which may be due to the inheritance of molecular regulators across mitosis (*Spencer et al., 2013*; *Mitchell et al., 2018*; *Yang et al., 2017*; *Barr et al., 2017*; *Arora et al., 2017*). By contrast, ancestral correlations in cycle length fade rapidly, often disappearing between grandmother and granddaughter cells, or already between mother and daughter cells.

Remarkably, however, the cycle lengths of cousin cells are found to be correlated, indicating that the grandmothers exert concealed effects through at least two generations. High intra-generational correlations in the face of weak ancestral correlations have been observed in cells as diverse as bacteria (*Powell, 1958*), cyanobacteria (*Yang et al., 2010*), lymphocytes (*Markham et al., 2010*) and mammalian cancer cells (*Staudte et al., 1984*; *Sandler et al., 2015*; *Chakrabarti et al., 2018*). The ubiquity of this puzzling phenomenon suggests that it may help reveal basic principles that control cell-cycle duration.

**\*For correspondence:**
t.hoefer@dkfz.de

[†]These authors contributed equally to this work

**Competing interests:** The authors declare that no competing interests exist.

Theoretical work has shown that more than one heritable factor is required to generate the observed cell-cycle correlations in T cell lineage trees, while the nature of these heritable factors has remained unclear (*Markham et al., 2010*). Stimulated by the idea of circadian gating of the cell cycle in cyanobacteria (*Mori et al., 1996*; *Yang et al., 2010*), recent comprehensive analyses of cell lineage trees across different species have proposed circadian clock control as a source of cell-cycle variability that can produce the observed high intra-generational correlations (*Sandler et al., 2015*; *Mosheiff et al., 2018*; *Martins et al., 2018*; *Py et al., 2019*); such a model also reproduced observed cycle correlations in colon cancer cells during chemotherapy (*Chakrabarti et al., 2018*). However, in proliferating mammalian cells in culture, the circadian clock has been found to be entrained by the cell cycle (*Bieler et al., 2014*; *Feillet et al., 2014*). Moreover, the circadian clock is strongly damped or even abrogated by oncogenes such as MYC (*Altman et al., 2015*; *Shostak et al., 2016*) yet MYC-driven cancer cells retain high intra-generational correlations (*Ryl et al., 2017*).

Ultimately, the cell cycle must coordinate growth and division in order to maintain a well-defined cell size over many generations. Yeast species have long served as model systems. Here, it is assumed that growth drives cell-cycle progression, although molecular mechanisms of size sensing remain controversial (*Facchetti et al., 2017*; *Schmoller and Skotheim, 2015*). By contrast, animal cells can grow very large without dividing (*Conlon and Raff, 2003*), and recent precise measurements suggest that growth control involves both modulation of growth rate and cell-cycle length (*Sung et al., 2013*; *Tzur et al., 2009*; *Cadart et al., 2018*; *Ginzberg et al., 2018*; *Liu et al., 2018*). A minimal requirement for maintaining cell size is that cells reach a critical size before dividing, which can be achieved by delaying S phase (*Shields et al., 1978*).

Here, we present a systematic approach to learning mechanisms from measured correlation patterns of cell cycles in deep lineage trees. First, we develop an unbiased statistical framework to identify the minimal model capable of accounting for our experimental data. We then propose a biological realization of this abstract model based on growth, inheritance and a size checkpoint, and experimentally test specific predictions of the biological model.

## Results

### Lineage trees exhibit extended intra-generational correlations

To study how far intra-generational cell-cycle correlations extend within cell pedigrees, we generated extensive lineage trees by imaging and tracking TET21N neuroblastoma cells for up to ten generations during exponential growth (*Figure 1A*, *Figure 1—video 1*, *Figure 1—source data 1* and *Figure 1—figure supplement 1A*). Autonomous cycling of these cells is controlled by ectopic expression of the *MYC*-family oncogene *MYCN*, overcoming the restriction point and thus mimicking the presence of mitogenic stimuli (*Ryl et al., 2017*). High *MYCN* also downregulated circadian clock genes (*Figure 1—figure supplement 2*). The distribution of cycle lengths (*Figure 1B* and *Figure 1—figure supplement 1B*) was constant throughout the experiment (*Figure 1C* and *Figure 1—figure supplement 1C*) and similar across lineages (*Figure 1—figure supplement 1D*), showing absence of experimental drift and of strong founder cell effects, respectively. To determine cycle-length correlations without censoring bias caused by finite observation time (*Figure 1—figure supplement 3A*; *Sandler et al., 2015*), we truncated all trees after the last generation completed by the vast majority (>95%) of lineages. The resulting trees were 5–7 generations deep, enabling us to reliably calculate Spearman rank correlations between relatives up to second cousins (*Figure 1D,E* and *Figure 1—figure supplement 3B*).

Cycle-length correlations of cells with their ancestors decreased rapidly with each generation (*Figure 1E*). However, the correlations increased again when moving down from ancestors along side-branches—from the grandmother toward the first cousins and also from the great-grandmother toward the second cousins (*Figure 1E*). The correlations among second cousins varied somewhat between replicates (we will show below that we can control these correlations experimentally by applying molecular perturbations). If cell-cycle length alone were inherited (e.g. by passing on regulators of the cell cycle to daughter cells), causing a correlation coefficient of $\rho_{\mathrm{md}}$ between mother and daughter cycle lengths, and sisters are correlated by $\rho_{\mathrm{ss}}$, then first and second cousins would be expected to have cycle length correlation $\rho_{\mathrm{ss}}\rho_{\mathrm{md}}^2$ and $\rho_{\mathrm{ss}}\rho_{\mathrm{md}}^4$, respectively (*Staudte et al., 1984*). The

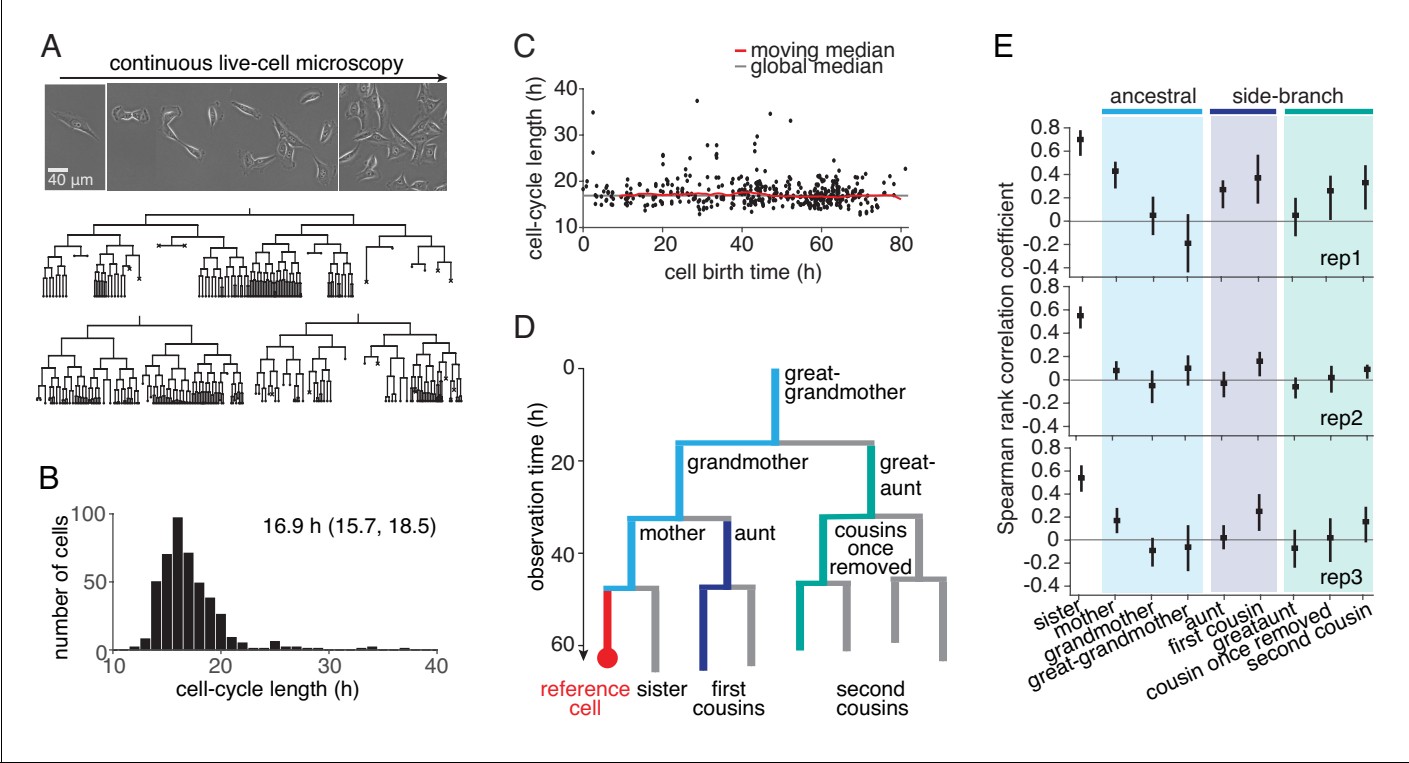

**Figure 1.** Cell-cycle lengths and their correlations captured by live-cell imaging. (A) Live-cell microscopy of neuroblastoma TET21N cell lineages. Sample trees shown with cells marked that were lost from observation (dot) or died (cross). (B) Distribution of cycle lengths, showing median length (and interquartile range). (C) Cycle length over cell birth time shows no trend over the duration of the experiment. (D) Lineage tree showing the relation of cells with a reference cell (red); ancestral lineage (light blue), first side-branch (dark blue) and second side branch (green). (E) Spearman rank correlations of cycle lengths between relatives (with bootstrap 95%-confidence bounds) of three independent microscopy experiments. Color code as in D. B and C show replicate rep3.

The online version of this article includes the following video, source data, and figure supplement(s) for figure 1:

**Source data 1.** Overview of all time-lapse experiments displayed in the manuscript.
**Source data 2.** Raw cell cycle data for lineage trees in TET21N replicates rep1-3.
**Figure supplement 1.** Temporal drift analysis of time-lapse imaging data.
**Figure supplement 2.** Expression of the circadian clock module depends on MYCN level.
**Figure supplement 3.** Censoring bias and spatial trend analysis.
**Figure 1—video 1.** Time-lapse movie of dividing TET21N cells (replicate rep3).
https://elifesciences.org/articles/51002#fig1video1

actually observed cousin correlations are much larger, confirming previous observations on first cousins as summarized in *Sandler et al. (2015)* and extending them to second cousins. This discrepancy between simple theoretical expectation and experimental data was not due to spatial inhomogeneity or temporal drift in the data (*Figure 1*; *Figure 1—figure supplement 3C-E*). Thus, the lineage trees show long-ranging intra-generational correlations that cannot be explained by the inheritance of cell-cycle length.

## Correlation patterns are explained by long-range memories of two antagonistic latent variables

We used these data to search for the minimal model of cell-cycle control that accounts for the observed correlation pattern of lineage trees (Materials and methods and Appendix 2). To be unbiased, we assumed that cycle length $\tau$ is controlled jointly by a yet unknown number $d$ of cellular quantities that are inherited from mother to daughter, $\boldsymbol{x} = (x_1, \ldots, x_d)^T$, such that $\tau = \tau(\sum_{l=1}^{d} \alpha_l x_l)$, with positive weights $\alpha$. We take $\boldsymbol{x}$ to be a Gaussian latent variable and, generalizing previous work (*Cowan and Staudte, 1986*), describe its inheritance by a generic model accounting for inter-

generational inheritance as follows: In any given cell $i$, $x^i$ is composed of an inherited component, determined by $x$ in the mother, and a cell-intrinsic component that is uncorrelated with the mother. The inherited component is specified by an inheritance matrix $\mathbf{A}$, such that the mean of $x^i$ conditioned on the mother's $x$ is $\langle x^i|x \rangle = \mathbf{A}x$ (*Figure 2A*). The cell-intrinsic component causes variations around this mean with covariance $\langle (x^i - \mathbf{A}x)(x^i - \mathbf{A}x)^T|x \rangle = \mathbf{I}$, where, with appropriate normalization of the latent variables, $\mathbf{I}$ is the unit matrix. Additional positive correlations in sister cells may arise due to inherited factors accumulated during, but not affecting, the mother's cycle (*Arora et al., 2017*; *Barr et al., 2017*; *Yang et al., 2017*); additional negative correlations may result from partitioning noise (*Sung et al., 2013*). These are captured by the cross-covariance between the intrinsic components in sister 1 and 2, $\langle (x^1 - \mathbf{A}x)(x^2 - \mathbf{A}x)^T|x \rangle = \gamma\mathbf{I}$. In total, $d(d+1)$ parameters can be

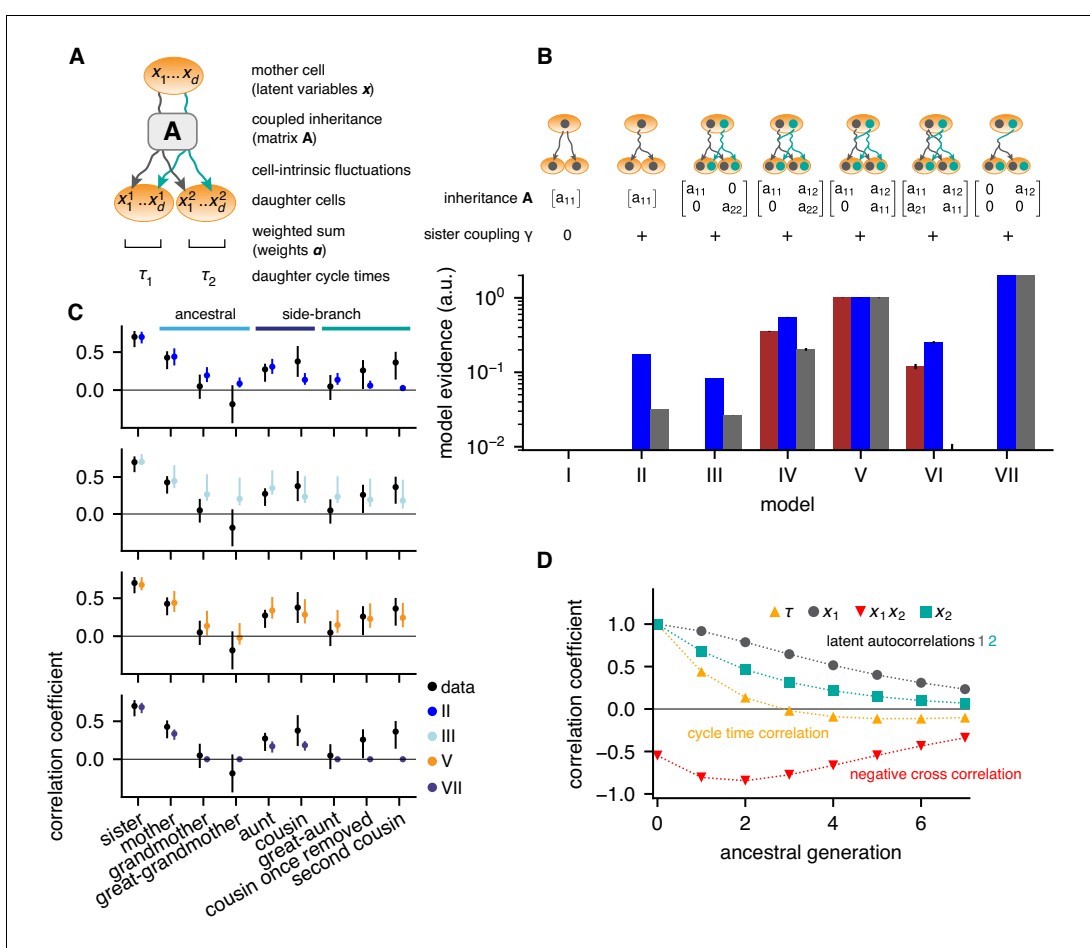

**Figure 2.** Bifurcating autoregressive inheritance models. (A) Coupled inheritance of $d$ Gaussian latent variables $x_l$ and cell-intrinsic fluctuations generate cycle lengths. (B) Relative model evidences calculated for $d = 1, 2$, for the indicated inheritance matrices $\mathbf{A} = [a_{lm}]$ and sister coupling $\gamma$. Although Model VII is the most parsimonious for replicates rep2 and rep3 (blue and gray bars), only Model V with unidirectionally coupled inheritance explains all data well, including rep1 (bordeaux bars). Error bars from Monte-Carlo integration. (C) Model fits for rep1. Single-variable inheritance (Model II) and pure cross-inheritance (VII) fails to generate strong intra-generational correlations; uncoupled inheritance (III) fails to generate low ancestral correlations; Model V fits the data best. Rank correlations of the data shown with bootstrap 95%-confidence bounds (black bars). Model prediction bands (colored bars) were generated from the range of the parameter sets with likelihood higher than 15% of the best fit, corresponding to a Gaussian 95% credible region. (D) Model V, best-fit ancestral autocorrelation functions, for cycle lengths $\tau$ and latent variables. Long-range memory in the latent variables is anticorrelated and masked in observed cycle times.

The online version of this article includes the following figure supplement(s) for figure 2:

**Figure supplement 1.** Gaussian model predictions of correlations for all three replicates.

adjusted to fit the correlation pattern of the lineage trees: the components $a_{lm}$ of the inheritance matrix $\mathbf{A}$, the weights $\alpha_l$ and the sister correlation $\gamma$. Together, these inheritance rules specify bifurcating first-order autoregressive (BAR) models for multiple latent variables governing cell-cycle duration.

To determine the most parsimonious BAR model supported by the experimental data, we employed a standard Bayesian model selection scheme. Selection is based on the Bayesian evidence, which rewards fit quality while naturally penalizing models of higher complexity (defined as being able to fit more diverse data sets; for details see Appendix 2, Evidence calculation). Specifically, we evaluated the likelihood of the measured lineage trees for a given BAR model, used it to compute the Bayesian evidence, and ranked BAR models accordingly (*Figure 2B*).

The simplest model that generated high intra-generational correlations was based on the independent inheritance of two latent variables (Model III; *Figure 2C*, cyan dots), whereas one-variable models failed to meet this criterion (Model II, *Figure 2C*, blue dots and Model I). However, Model III consistently overestimated ancestral correlations and hence its relative evidence was low (<10% for all data sets). To allow additional degrees of freedom, we accounted for interactions of latent variables. The most general two-variable model with bidirectional interactions (Model VI), overfitted the experimental data and consequently had low evidence. The models best supported by the data had unidirectional coupling, such that $x_2$ in the mother negatively influenced $x_1$ inherited by the daughters, that is with $a_{12}<0$ and $a_{21}=0$ (*Figure 2B*, Models IV, V and VII). Among these, Model VII, with a single inheritance parameter $a_{12}$, is simplest, but was not compatible with experimental replicate rep1 as it could not generate second-cousin correlations (*Figure 2B*,C). Both Models IV and V were compatible with all replicates; however, Model V with only one self-inheritance parameter for both variables ($a_{11}=a_{22}>0$) was preferred (Model V, *Figure 2B,C*, orange dots). Model V produced a remarkable inheritance pattern (*Figure 2D*): Individually, both latent variables had long-ranging memories, with ~50% decay over 2–3 generations. However, the negative unidirectional coupling cross-correlated the variables negatively along an ancestral line, resulting in cycle-length correlations that essentially vanished after one generation. Nevertheless, strong intra-generational correlations were reproduced by the model due to long-range memories of latent variables together with positive sister-cell correlations ($\gamma>0$). We conclude that the coexistence of rapidly decaying ancestral correlations and extended intra-generational correlations can be explained by the inheritance of two latent variables, one of which inhibits the other.

## Cell size and speed of cell-cycle progression are antagonistic heritable variables

During symmetric cell division, both cell size and regulators of cell-cycle progression are passed on from the mother to the daughter cells (*Spencer et al., 2013*; *Yang et al., 2017*; *Arora et al., 2017*; *Barr et al., 2017*). We now show that simple and generic inheritance rules for these two variables provide a physical realization for BAR Model V.

To divide, cells need to both grow to a minimum size (*Shields et al., 1978*) and receive license to progress through the cell cycle from the regulatory machinery (*Novak et al., 2007*). Indeed, growth and cell-cycle progression can be separately manipulated experimentally in mammalian cells (*Fingar et al., 2002*). In particular, cells continue to grow in size when regulatory license is withheld, for example in the absence of mitogens, and growth is not otherwise constrained, for example by mechanical force or growth inhibitors (*Fingar et al., 2002*; *Conlon and Raff, 2003*).

While growth and cell-cycle progression are separable and heritable processes, they also interact. At the very least, the length of the cell cycle needs to ensure that cells grow to a sufficient size for division. This interaction alone implies an effect of one inherited variable, cycle progression, on the other, cell growth, that anti-correlates subsequent cell cycles (as required by BAR model V): If a delayed regulatory license prolongs the mother's cell cycle, it will grow large. By size inheritance, its daughters will be large at birth, reach a size sufficient for division quickly and hence may have shorter cell cycles. Thus, despite inheritance of growth and cell-cycle regulators mothers and daughters may have very different cycle lengths due to this interaction.

Based on these ideas, we formulated a simple quantitative model of growth and cell-cycle progression on cell lineage trees. We introduced the variables 'cell size' $s$, measuring metabolic, enzymatic and structural resources accumulated during growth, and $p$, characterizing the progression of the cell-cycle regulatory machinery. Unlike the latent variables of the BAR model $x_1$ and $x_2$, their

mechanistic counterparts $s$ and $p$, respectively, are governed by rules reflecting basic biological mechanisms (*Figure 3A*, Appendix 3). Size $s$ grows exponentially and is divided equally between the daughters upon division. We found that under some experimental conditions generating long cell cycles (downregulation of *MYCN*, see below), stable cell size distributions required feedback regulation of growth rate, as seen experimentally (*Sung et al., 2013*; *Tzur et al., 2009*); we implemented this as a logistic limitation of growth rate at large sizes for these conditions. The progression variable $p$ determines the time taken for the regulatory machinery to complete the cell cycle, which is controlled by the balance of activators and inhibitors of cyclin-dependent kinases. These regulators are inherited across mitosis (*Spencer et al., 2013*; *Yang et al., 2017*; *Arora et al., 2017*; *Barr et al., 2017*) and hence the value of $p$ is passed on to both daughter cells with some noise. Cells divide when they have exceeded a critical size, requiring time $\tau_g$, and the regulatory machinery has

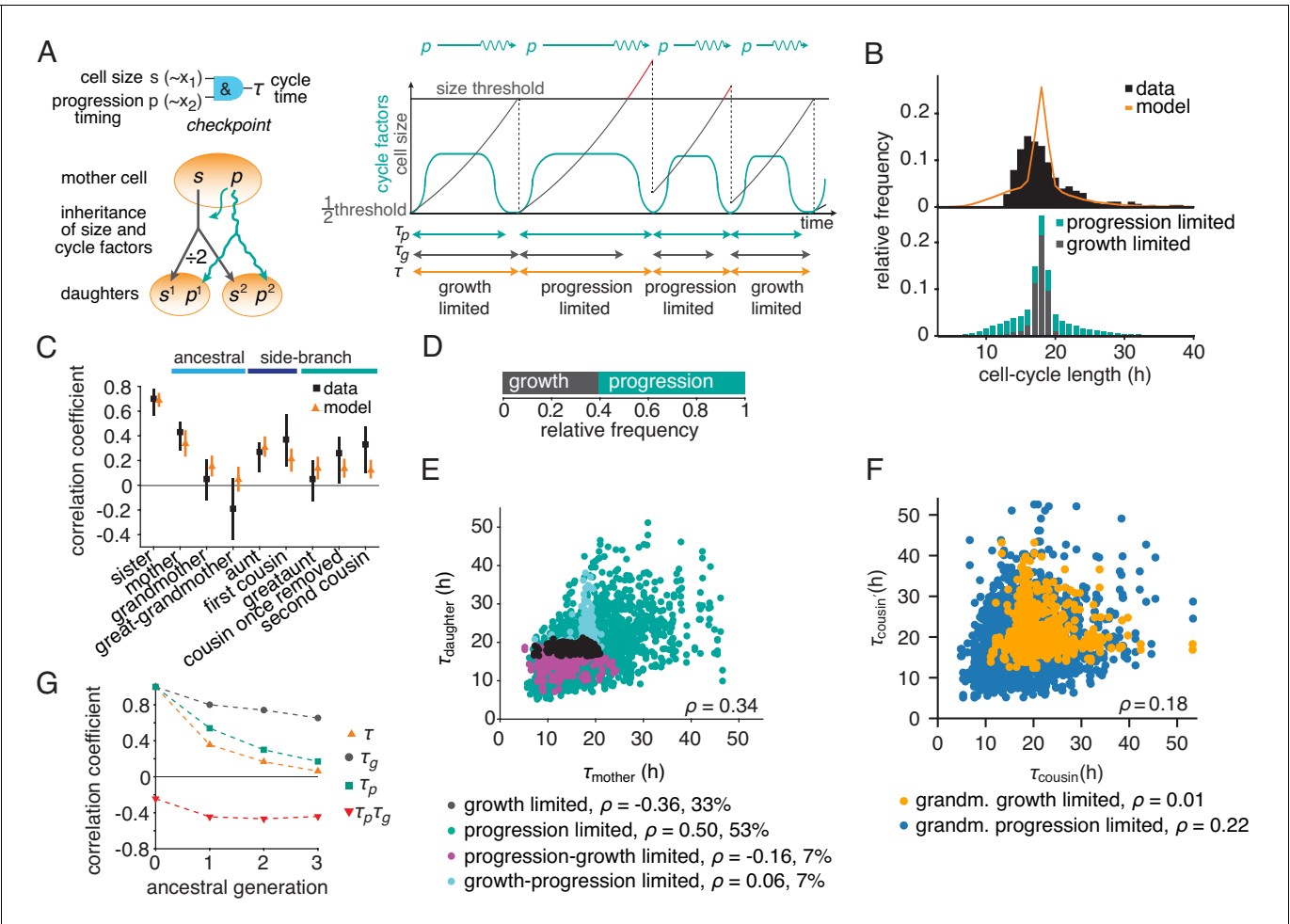

**Figure 3.** The growth-progression model. (A) Scheme of the growth-progression model with heritable variables relating to cell size $s$ and cycle progression timing $p$. (B) Measured and simulated cell-cycle length distributions (upper). Model distribution resolved by the division-limiting process (lower). (C) Measured and modeled correlation pattern with Spearman rank correlation coefficient and bootstrap 95%-confidence bounds. (D) Proportion of simulated cells limited by growth or progression. (E) Correlation of simulated mother-daughter cycle lengths colored by their division limitation: both by $\tau_g$ (black), both by $\tau_p$ (green), mother $\tau_p$ – daughter $\tau_g$ (magenta), mother $\tau_g$ – daughter $\tau_p$ (cyan). Percentage of cells in each subgroup and their correlation coefficients are shown. (F) Correlation of simulated cousin-cousin cycle length colored by the limitation of the common grandmother: by $\tau_g$ (orange) or $\tau_p$ (blue). (G) Autocorrelations along ancestral line of cycle length $\tau$, growth time $\tau_g$ and the progression time $\tau_p$, and the cross-correlation $\tau_p\tau_g$.

The online version of this article includes the following source data and figure supplement(s) for figure 3:

**Source data 1.** Best-fit parameter values of the growth-progression model for all experiments shown, obtained from ABC-simulations.
**Figure supplement 1.** Parameterized growth-progression model generates long-range memory.

progressed through the cycle, which takes an approximately log-normally distributed time (*Ryl et al., 2017*; *Mitchell et al., 2018*) modeled as $\tau_p = \exp(p)$. Hence the cycle length is $\tau = \max(\tau_g, \tau_p)$. Apart from requiring a minimum cell size for division, the growth-progression model does not implement a drive of the cell cycle by growth and thus allows cells to grow large during long cell cycles. By this mechanism, the cell size variable $s$ is influenced by cycle progression, analogous to the BAR variable $x_1$. By contrast, the progression variable $p$ is not influenced by cell size, analogous to the variable $x_2$ in the BAR model.

We fitted this model to the measured lineage trees by Approximate Bayesian Computation (*Figure 3—figure supplement 1A* and *Figure 3—source data 1*). The parameterized model yielded a stationary cell size distribution (*Figure 3—figure supplement 1B*) and reproduced the cycle-length distribution (*Figure 3B* and *Figure 3—figure supplement 1C*) as well as the ancestral and intra-generational correlations (*Figure 3C* and *Figure 3—figure supplement 1D*). Thus, the dynamics of cell growth and cell-cycle progression, coupled only through a minimum-size requirement, account for the intricate cycle-length patterns in lineage trees.

To gain intuition on the inheritance patterns of cycle length, we first considered ancestral correlations, focusing on mother-daughter pairs. Individual cell cycles in the model are either growth-limited, that is division happens upon reaching the minimum size, or progression-limited, that is the cell grows beyond the minimum size until the cycle is completed (*Figure 3D* and *Figure 3—figure supplement 1E*). If both mother and daughter are progression-limited (i.e., the threshold size is exceeded by both), their cycles are positively correlated (*Figure 3E*, green dots). As in this case size inheritance is inconsequential, this positive correlation is explained by the inheritance of the cell-cycle progression variable $p$ alone. By contrast, all mother-daughter pairs that involve at least one growth limitation show near-zero (*Figure 3E*, cyan dots) or negative correlations (*Figure 3E*, magenta and black dots). This pattern is explained by the anti-correlating effect that daughters of longer-lived and hence larger mother cells require on average shorter times to reach the size threshold. Next, we considered intra-generational correlations, focusing on first cousins (*Figure 3F*). While cousins are positively correlated overall, this correlation is carried specifically by cousins that descend from a grandmother with a progression-limited cell cycle (*Figure 3F*, blue dots), whereas cousins stemming from a growth-limited grandmother are hardly correlated (*Figure 3F*, orange dots). Since progression-limited cells can grow large, this observation indicates that cousin correlations are mediated by inheritance of excess size, as is confirmed by conditioning cousin correlations on grandmother size (*Figure 3—figure supplement 1F*). Size inheritance over several generations is also evident in the autocorrelation of the time required to grow to minimum size, $\tau_g$ (*Figure 3G* and *Figure 3—figure supplement 1G*, black dots). The autocorrelation of the progression time $\tau_p$ is also positive (but less long-ranging; *Figure 3G* and *Figure 3—figure supplement 1G*, green squares), while the negative interaction with growth is reflected in the negative cross-correlation (*Figure 3G* and *Figure 3—figure supplement 1G*, red triangles). In sum, the long-range memories of cell-cycle progression and cell growth are masked by negative coupling of these processes, causing rapid decay of cell-cycle length correlations along ancestral lines (*Figure 3G* and *Figure 3—figure supplement 1G*, orange triangles). These inheritance characteristics of the growth-progression model mirror those of BAR model V (see *Figure 2D*).

## Effects of molecular perturbations on cell-cycle correlations are correctly predicted by the model

If the growth-progression model captures the key determinants of the cell-cycle patterns in lineage trees, it should be experimentally testable by separately perturbing growth versus cell-cycle progression. We first derived model predictions for these experiments. Intuitively, if growth limitation were abolished by slowing cell-cycle progression, only progression would be inherited and therefore mother-daughter correlations of cycle time could no longer be masked. As a result, we expect intra-generational correlations to be reduced relative to ancestral correlations when inhibiting cycle progression; conversely, inhibiting growth (and thereby increasing growth limitation) should raise them. Indeed, using the model to simulate perturbation experiments (*Figure 3—source data 1*), we found that growth inhibition increased cousin correlations relative to mother-daughter correlations, whereas slowing cell-cycle progression decreased these correlations (*Figure 4A*).

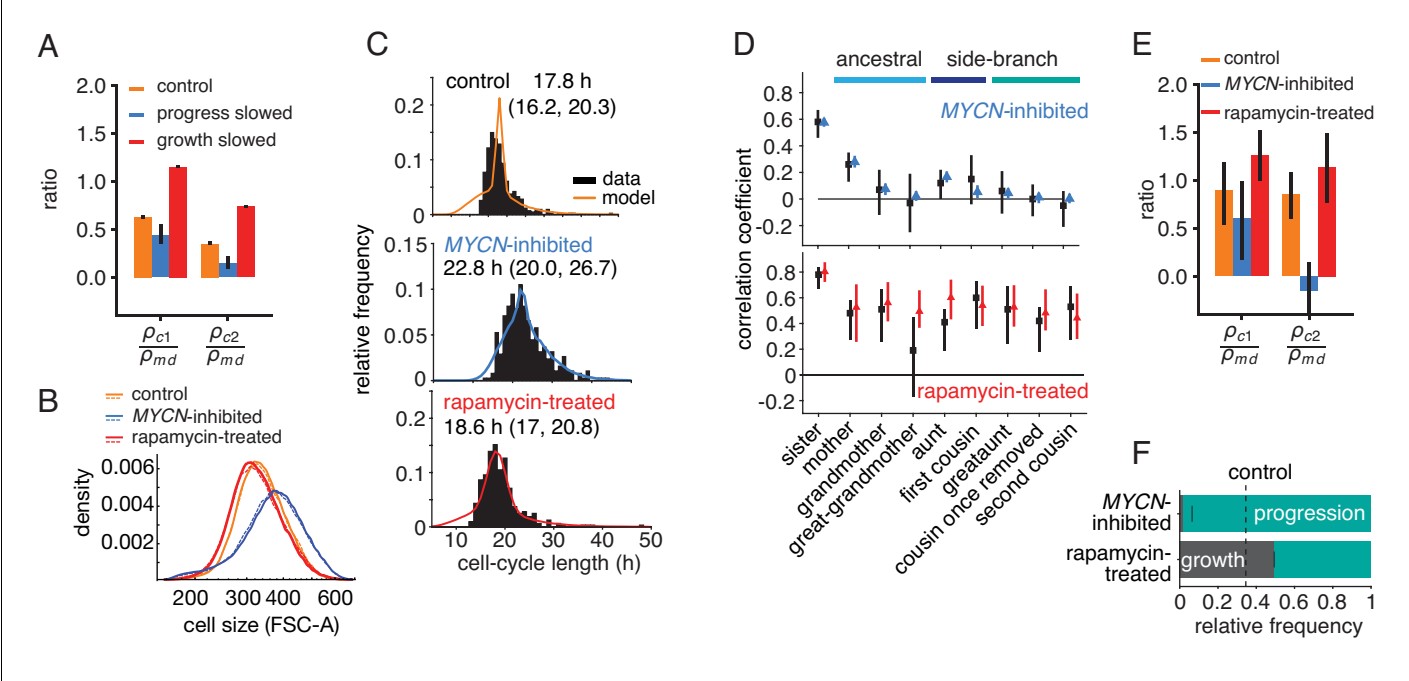

**Figure 4.** Targeted perturbation of growth and cell-cycle progression. (A) Predictions for changes in the ratio $\rho_c/\rho_{md}$ of cousin to mother-daughter correlations, when slowing growth or cycle progress compared to the best-fit parameters (control). $\rho_{c1}$ = first cousins, $\rho_{c2}$ = second cousins. (B–F) Experimental perturbations of cycle progress and growth by *MYCN* inhibition and rapamycin treatment, respectively. (B) Cell size distribution. Areal forward scatter measured experimentally by flow cytometry for control *high*-MYCN, *MYCN*-inhibited and rapamycin-treated (40 nM) TET21N neuroblastoma cells; shown are two biological replicates, indicated by solid and dashed lines, that were measured with the same FACS settings. (C) Measured and best-fit model cycle length distributions. Median and interquartile range are indicated. (D) Measured (black) and best-fit correlation pattern of *MYCN*-inhibited and rapamycin-treated cells with Spearman rank correlation coefficient and 95%-confidence bounds. (E) Measured cousin/ mother daughter correlation ratios. (F) Proportion of simulated cells limited by growth or progression, using best-fit parameters for *MYCN* inhibition or rapamycin-treatment.

The online version of this article includes the following source data and figure supplement(s) for figure 4:

**Source data 1.** Raw cell cycle data for lineage trees in perturbed TET21N replicates -myc1-2 and rap1-2.
**Figure supplement 1.** Growth-progression and BAR models fitted to perturbation data.
**Figure supplement 2.** Logistic growth-progression model fitted to all control, rapamycin-treated and embryonic stem cell datasets.

To test this prediction, we slowed cell-cycle progression experimentally by reducing MYCN, exploiting the doxycycline-tunable *MYCN* gene integrated in the TET21N cells. Cells grew to larger average size (*Figure 4B*, blue lines) over longer and more variable cell cycles (*Figure 4C*). These data show that lowering MYCN slowed cell-cycle progression while allowing considerable cell growth. Further consistent with this phenotype, expression of mTOR, a central regulator of metabolism and growth (*Fingar et al., 2002*), was not lowered (*Figure 4—figure supplement 1A*). In a separate experiment, we inhibited cell growth by applying the mTOR inhibitor rapamycin, which reduced cell size by a small but reproducible amount (*Figure 4B*, red lines). This treatment also lengthened the cell cycle slightly (*Figure 4C*) but without changing MYCN protein levels (*Figure 4— figure supplement 1B*). Thus, lowering MYCN and inhibiting mTOR are orthogonal perturbations that act on cell-cycle progression and cell growth, respectively. As predicted by the growth-progression model, these perturbations resulted in markedly different cycle-length correlation patterns within lineage trees (*Figure 4D,E* and *Figure 4—figure supplement 1C,D*): Lowering MYCN decreased intra-generational correlation and, in particular, removed second-cousin correlations. By contrast, rapamycin treatment strongly increased intra-generational correlations and caused ancestral correlations to decline only weakly. Collectively, these findings support the growth-progression model of cell-cycle regulation.

## Cousin correlations reflect active cell-size checkpoint

We then asked whether the cycle-length correlation patterns experimentally observed in lineage trees contain information about the underlying regulation. To this end, we fitted the BAR and growth-progression models to MYCN and rapamycin perturbation data. We again obtained good agreement with the data (*Figure 4C,D* and *Figure 4—figure supplement 1C–E,G–J*). *MYCN* knockdown cells grew larger; to obtain a stable cell-size distribution for the corresponding model fits, we implemented logistic regulation of growth rate at large cell sizes. When we applied, for comparison, this regulation to the control and rapamycin treatment data, the model fits (*Figure 4—figure supplement 2*) were not noticeably affected compared to purely exponential growth. This result indicates that growth rate regulation affecting large cells, as found experimentally (*Ginzberg et al., 2018*), is compatible with long-range intra-generational correlations in cell-cycle length.

In terms of fit parameters of the model, *MYCN* inhibition caused considerable slowing of cycle progression and also a moderate decrease in growth rate (*Figure 3—figure supplement 1A*, parameters $\mu$ and $k$, respectively). As a result, the vast majority of cell cycles were progression-limited (*Figure 4F* and *Figure 4—figure supplement 1F*). For the rapamycin-treated cells, we estimated growth rates that were lower than for the control experiments on average, as expected (*Figure 3—figure supplement 1A*, parameter $k$). Also, correlations in cell-cycle progression increased in mother-daughter and sister pairs (parameters $a$ and $\gamma$, respectively). This is consistent with rapamycin inhibiting mTOR and hence growth, but not affecting drivers of the cell cycle, ERK and PI3K (*Adlung et al., 2017*), since then lengthening of the cell cycle due to slower growth may allow prolonged degradation of cell-cycle inhibitors, which would increase inheritance of cell-cycle length (*Smith and Martin, 1973*). Taken together, rapamycin treatment increased the fraction of growth-limited cell cycles (*Figure 4F* and *Figure 4—figure supplement 1F*) and inheritance of cell-cycle progression speed, thus causing increased ancestral and intra-generational correlations.

We hypothesized that growth may be limiting primarily for rapidly proliferating cell types, even without specific growth inhibition. We analyzed time-lapse microscopy data of non-transformed mouse embryonic stem cells (*Filipczyk et al., 2015*) that proliferate much faster than the neuroblastoma cells (*Figure 5A* and *Figure 5—figure supplement 1A*). Side-branch correlations of cycle length were again large (*Figure 5B Figure 5—figure supplement 1B*), as seen in the previous data except for the *MYCN*-inhibited cells. Interestingly, the strength of the intra-generational correlations was most similar to the much more slowly dividing rapamycin-treated cells (cf. *Figure 4D*). As before, the BAR model required two negatively coupled variables to account for these data (*Figure 5C*, *Figure 5—figure supplement 1D,E*). Fitting the growth-progression model to the data (*Figure 5A,B*), we found that the majority (~60%) of cell cycles were limited by growth (*Figure 5D*, *Figure 5—figure supplement 1C*), indicating that cycle length of fast proliferating mammalian cells is, to a large extent, controlled by growth.

## Discussion

Here, we showed that the seemingly paradoxical pattern of cell-cycle lengths in lineage trees, with rapidly decaying ancestral and long-range intra-generational correlations, can be accounted for by the inheritance of two types of quantities: resources accumulated during the cell cycle (cell 'size') and regulators governing the speed of cell-cycle progression. The fact that these are fundamental processes in dividing cells may help explain the ubiquity of the paradoxical cell-cycle pattern. Targeted experimental perturbations of cell growth and cell-cycle progression support our model.

As an alternative mechanism underlying the observed cell-cycle variability and cousin correlations, modulation of the cell cycle by the circadian clock has been suggested, with strongest experimental evidence to date for cyanobacteria (*Sandler et al., 2015*; *Mosheiff et al., 2018*; *Py et al., 2019*). Unlike cyanobacteria, proliferating mammalian cells show entrainment of the circadian clock to the cell cycle with periods well below 24 hr (*Bieler et al., 2014*; *Feillet et al., 2014*). Whether in this setting the circadian clock could still influence cell-cycle correlations remains to be studied.

Our proposed mechanism for cycle length correlations was motivated by Bayesian model selection. All data sets which displayed long-range correlation patterns (control replicate rep1 and rapamycin treatment of neuroblastoma cells as well as ESCs) selected a BAR model featuring long-range inheritance of two memory variables, which is masked in mother-daughter pairs by an anticorrelating interaction between them. The growth-progression model shares these essential features. Size

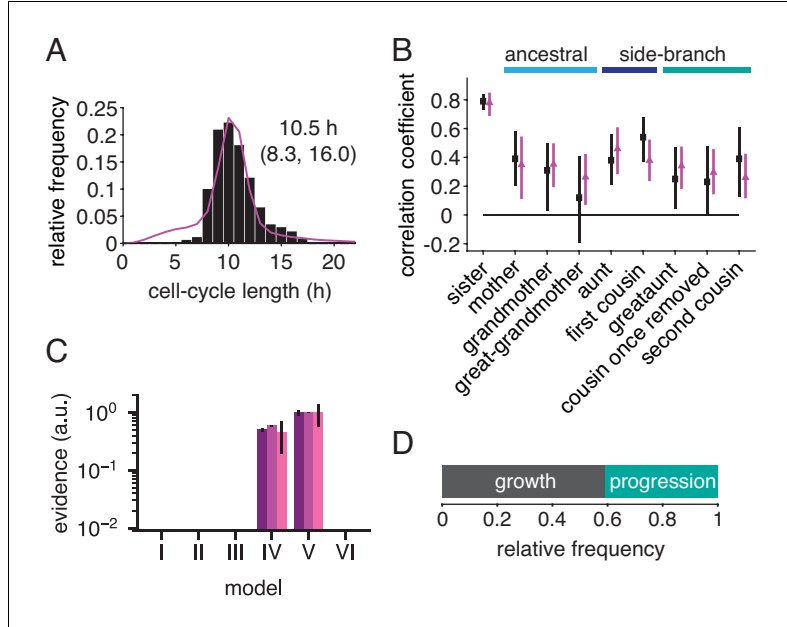

**Figure 5.** Rapid cell cycle of embryonic stem cells are frequently growth-limited. (**A**) Cycle length distribution of data (black) and growth-progression model (purple). (**B**) Measured (black) and modeled (purple) correlation pattern using the growth-progression model. (**C**) Model evidences of the BAR model, version numbering as in *Figure 2B*. (**D**) Proportion of simulated cells limited by growth or progression. Data from *Filipczyk et al. (2015)* reanalyzed for cell-cycle duration.

The online version of this article includes the following figure supplement(s) for figure 5:

**Figure supplement 1.** Growth-progression and BAR models fitted to embryonic stem cell data.

---

inheritance is predicted to be particularly long-range by the fits of the growth-progression model to the experimental data. This memory emerges because growth does not force cell-cycle progression, allowing cells to grow large in progression-limited cycles and then pass on size over several generations. Such weak coupling between concurrent processes echoes recent work in *E. coli* on timing of cell division (*Micali et al., 2018*) and on the existence of two parallel adder processes for replication and division (*Witz et al., 2019*). Of note, the latter paper also utilizes a statistical framework for systematic model selection against experimentally observed correlation patterns akin to our approach.

Our findings raise the question of the mechanistic modes by which cells coordinate cell cycle and growth, which has been a long-standing problem; see *Shields et al. (1978)* for historical references and *Ho et al. (2018)* for a recent review. Recent work on this problem has shown the existence of negative feedback of cell size on growth rate (*Ginzberg et al., 2018*). We have found that our results remain robust when implementing a simple form of such a feedback (logistic dependence of growth rate on size) in the growth-progression model. Another recent study has shown that growth of many mammalian cells during the cell cycle adds a volume that only weakly increases with cell volume at birth (termed near-adder behavior, *Cadart et al., 2018*). This behavior appears to be caused by a combination of growth-rate regulation and cell-size effects on cell-cycle progression. We expect that factoring in the cell-cycle length correlations studied here will help uncover the mechanistic details of cell size regulation. Refining our model in this direction may also help capture yet more detail of the correlation structure, such as the apparent increasing trend from aunt to first cousins and greataunt to second cousins. Moreover, we envisage that our inference approach could be extended to include finely resolved data on cell-cycle phases (*Chao et al., 2019*) as well as multiple cell fates via asymmetric division and differentiation (*Duffy et al., 2012*).

## Materials and methods

### Experimental methods

*MYCN*-tunable mammalian neuroblastoma SH-EP TET21N (RRID:CVCL_9812) cells were cultured as in *Lutz et al. (1996)*. These TET21N cells were obtained from Dr. Frank Westermann (German Cancer Research Center, DKFZ) whose lab generated this line. This cell line is regularly authenticated by an in-house DKFZ service using STR profiling. Mycoplasma contamination testing was negative. *MYCN* and mTOR were inhibited using 1 µg/ml doxycycline or 20–40 nM rapamycin (Calbiochem, 553210–100 UG), respectively. Cells were grown on ibidi µ-slides and phase contrast images (Nikon Ti-E) acquired every 6–15 min for up to 7 days under controlled growth conditions. The presented data consists of independent biological and technical replicates with n = 3 for untreated TET21N cells, n = 2 for *MYCN*-inhibited and rapamycin-treated cells. Cells were tracked in Fiji (version 1.48d) using the tracking plugin MTrackJ (*Meijering et al., 2012*). For flow cytometry, cells were stained with MYCN primary antibody (Santa Cruz Biotechnology, Cat# sc-53993; RRID:AB_831602), secondary fluorescence-conjugated antibody goat anti-mouse Alexa Fluor 488 IgG (Life Technologies Cat#A-11001; RRID:AB_2534069) and measured on a Miltenyi VYB MACSQuant Analyser. See Appendix 1 for details.

### Data analysis

MATLAB (R2016b) was used for all data analyses. Correlations represent Spearman rank correlations or, for the BAR model, Pearson correlation coefficients between the Gaussian-transformed cycle times. The difference between these two methods was far smaller than the experimental error. Error bounds were estimated by bootstrap re-sampling on the level of lineage trees. Censoring bias was avoided by truncating lineage trees after the last generation completed by all lineages within the experiment (see e.g. *Sandler et al., 2015* and Appendix 1), truncating the trees to 7, 6 and 5 generations for the three *MYCN* amplified experiments. *MYCN*-inhibited and rapamycin-treated trees were five generations deep.

### Bifurcating autoregressive (BAR) model

We constructed BAR models of cell-cycle inheritance, as described in detail in Appendix 2. Briefly, the cell state is determined by a vector of Gaussian (latent) variables which are inherited from the mother to the daughter cells by a linear map plus a cell-intrinsic noise term, which is correlated between daughters. The model is thus a Gaussian latent-variable model, where inheritance takes the form of an autoregressive vector-AR(1) process defined on a lineage tree. The cycle time is then calculated by an data-derived (approximately exponential) function of a weighted sum of the cellular state. We calculated whole-lineage tree log-likelihood functions analytically and used them to evaluate Bayesian Evidences (Bayes factors) that quantify the relative support from the data for various model variants.

### The growth-progression model

Cell-cycle progression is modeled by a fluctuating, centered Gaussian heritable variable $q$, analogous to version II of the BAR model. Variables were scaled and shifted, $p = \sigma_p q + \mu$, yielding log-normal progression durations $\tau_p = \exp(p)$. Size accumulation was modeled by exponential growth or for *MYCN* inhibition logistic growth. The normalized critical cell size $s_{\text{th}}$ fluctuates slightly and independently in each cell as $s_{\text{th}} = 1 + \zeta$ with $\zeta \sim \mathcal{N}(0, \sigma_g^2)$. The growth-progression model was implemented in Matlab (R2016b), R (3.4.3) and OCaml (4.06) and 30 trees of 7 generations simulated, corresponding to the experimental dataset sizes. The simulation was repeated 100 times to generate confidence bounds. Parameters were fitted using Approximate Bayesian Computation independently for each dataset. See Appendix 3 for details.

## Acknowledgements

We thank Frank Westermann and Tatjana Ryl for TET21N cells and laboratory setup; Timm Schroeder, Fabian Theis and Carsten Marr for embryonic stem cell data and discussions; Alessandro Greco for differential gene expression analysis and all members of the Höfer group for discussions.

Grant support to TH from BMBF (MYC-NET 0316076A and Sysmed-NB 01Z × 1307), EU (FP7-HEALTH-2010 ASSET 259348), BMBF and EU (EraCoSysMed OPTIMIZE-NB 031L0087A), as well as DKFZ core funding are gratefully acknowledged; TH is a member of CellNetworks.

## Additional information

### Funding

| Funder | Grant reference number | Author |
|---|---|---|
| Bundesministerium für Bildung und Forschung | 0316076A | Thomas Höfer |
| Bundesministerium für Bildung und Forschung | 01ZX1307 | Thomas Höfer |
| Bundesministerium für Bildung und Forschung | 031L0087A | Thomas Höfer |
| Seventh Framework Programme | FP7-HEALTH-2010 ASSET 259348 | Thomas Höfer |

The funders had no role in study design, data collection and interpretation, or the decision to submit the work for publication.

### Author contributions

Erika E Kuchen, Conceptualization, Data curation, Software, Formal analysis, Validation, Investigation, Visualization, Methodology, Writing - original draft, Writing - review and editing; Nils B Becker, Conceptualization, Software, Formal analysis, Investigation, Visualization, Methodology, Writing - original draft, Writing - review and editing; Nina Claudino, Data curation, Investigation, Writing - review and editing; Thomas Höfer, Conceptualization, Supervision, Funding acquisition, Writing - original draft, Writing - review and editing

### Author ORCIDs

Nils B Becker (ID) https://orcid.org/0000-0002-7490-6425
Thomas Höfer (ID) https://orcid.org/0000-0003-3560-8780

### Decision letter and Author response

Decision letter https://doi.org/10.7554/eLife.51002.sa1
Author response https://doi.org/10.7554/eLife.51002.sa2

## Additional files

### Supplementary files

- Supplementary file 1. Key resources table.
- Transparent reporting form

### Data availability

Data generated or analysed during this study are included in the manuscript and supporting files. Source data files have been provided for Figures 1 and 4.

The following previously published dataset was used:

| Author(s) | Year | Dataset title | Dataset URL | Database and Identifier |
|---|---|---|---|---|
| Ryl T | 2017 | RNA-Seq of SHEP TET21N cells upon Doxorubicin treatment | https://www.ncbi.nlm.nih.gov/geo/query/acc.cgi?acc=GSE98274 | NCBI Gene Expression Omnibus, GSE98274 |

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

## Appendix 1

## Experimental methods

### Cell culturing and treatment

*MYCN*-tunable mammalian neuroblastoma SH-EP TET21N (TET21N, RRID: CVCL_9812) (*Lutz et al., 1996*) cells were cultured in RPMI 1640 medium supplemented with 10% fetal calf serum and 1% penicillin/streptomycin at 37°C 5% $CO_2$ and 88% humidity. Versene was used for harvesting. TET21N cells were originally isolated from a female patient. Cell lines are authenticated by the German Cancer Research Center in house facility every half-year. *MYCN*-inhibited populations were established by incubating cells with 1 µg/ml doxycycline for 48–72 hr prior to further analysis. Growth-inhibited populations were generated by treating cells with the mTOR inhibitor rapamycin (Calbiochem, 553210–100 UG) at 20 nM or 40 nM rapamycin dissolved in DMSO. Cells were treated with rapamycin or the same concentration of DMSO for 72 hr prior to harvesting for flow cytometry or live-cell microscopy.

### Live-cell microscopy

$10^3$ cells were grown on 8-well ibidi µ-slides coated with collagen IV (Cat# 80822) in RPMI 1640 medium and imaged every 6–15 min for up to 7 days under controlled growth conditions at 37°C, 5% $CO_2$ and 80% humidity (Pecon incubator P). Growth media was changed every 2–3 days. Phase contrast images were acquired with an inverted widefield microscope (Nikon Ti-E) using an EMCCD camera (Andor iXON3 885) and a 10x (CFI Planfluor DL-10x, NA 0.3) or 20x lense (CFI Plan Apochromat DM 20x, NA 0.75). Cells were tracked in Fiji (version 1.48d) using the manual tracking plugin MTrackJ (*Meijering et al., 2012*). The presented imaging data consists of independent biological and technical replicates with n = 3 for untreated TET21N cells, n = 2 for *MYCN*-inhibited cells and n = 2 for rapamycin-treated cells.

### Flow cytometry and antibody staining

$10^6$ cells were fixed with 4% paraformaldehyde for 15 min at room temperature. Permeabilization was performed in 90% ice-cold methanol for at least 24 hr at $-20$°C. Cells were washed in staining buffer (1% BSA, 0.1% TritonX in PBS), and incubated with 0.5–1 µg per sample of MYCN primary antibody (Santa Cruz Biotechnology, Cat# sc-53993; RRID:AB_831602) for 1 hr at room temperature. Cells were washed 3x with staining buffer and incubated with a secondary flouresence-conjugated antibody, goat anti-mouse Alexa Fluor 488 IgG (Life Technologies Cat#A-11001; RRID:AB_2534069), again for 1 hr at room temperature. Cells were washed 3x with staining buffer and DNA content staining was performed with FxCycle Violet Stain (Thermo Fischer Scientific). A Miltenyi VYB MACSQuant Analyser was used for measurements and data was analysed using FlowJo software.

### Transcriptomics

mTOR mRNA expression data in TET21N cells under control and *MYCN*-inhibited conditions was obtained from RNA-Seq measurements by *Ryl et al. (2017)* deposited at GEO (GSE98274).

### Data analysis

MATLAB (R2016b) was used for all data analysis steps.

## Correlation coefficients

Correlation patterns of lineage trees of data and the growth-progression model represent Spearman rank correlations. For the bifurcating autoregressive (BAR) model, Pearson correlation coefficients are calculated between the Gaussian-transformed cycle times. These are the same as Gaussian rank correlations between cycle times. In practice, the differences between Gaussian rank correlations and Spearman rank correlations were much smaller than experimental error bounds on either, so we did not differentiate between the measures. Confidence bounds on correlation coefficients were estimated by bootstrap resampling on the level of lineage trees; resampling on the level of individual pairs of related cells would neglect the correlations between cells of the same tree and thus underestimate variability. Trees that did not contain information on the cell pair under analysis were removed prior to bootstrapping. At each bootstrap repeat, family trees were randomly drawn with replacement up to the number of trees in the original dataset. From the resulting bootstrap sample all cell pairs were used to calculate the sample correlation coefficient. This process was repeated 10,000 times. From the resulting distribution of correlation coefficients the 95% quantiles were used as confidence bounds.

## Exponential growth

For each experiment, an exponential growth model of the form $N_t = N_0 \exp(kt)$ was fitted to cell counts over time by performing a maximum likelihood estimation using the trust-region algorithm in MATLAB.

## Cell-cycle length distribution over time

At each time point, the moving-window median was calculated from all cells born within a window of 10 hr before or after this point.

## Randomization

All cells within a dataset were randomly paired with each other and correlation of the resulting sample calculated. This procedure was repeated 10,000 times. From the resulting distribution the mean and the 95% quantiles are given.

## Censoring

Censoring bias resulting from a finite observation time can lead to an overrepresentation of faster cells. To demonstrate this effect, we generated trees using a toy model with independent normally distributed cycle lengths. The trees were truncated at various total observation times and Spearman rank correlation coefficients between all related cells within the observation time window were sampled. As *Figure 1—figure supplement 3A* shows, short observation times strongly distort the sampled mother-daughter and sibling correlations away from their true value 0. The same basic effect persists for more distant relationships and can be further enhanced if cycle lengths are inherited. This censoring bias can be avoided by truncating lineage trees not after a given observation time, but after the last generation completed by all lineages within the experiment, uniformly over all trees (see *e.g.* *Sandler et al., 2015*). In this way slower and faster-cycling lineages are represented equally. Because some cells were inevitably lost from the field of view by migration, and a small percentage of cells showed extremely slow cycles, such a strict cut-off was unfeasible in our experiments. We assumed that cell loss by migration is not correlated to cycle length, so migrating cells were not counted as missing from otherwise complete trees. Within the remaining tree, we then determined the last generation to be included in our analysis by the following procedure: We first counted the number of cells $n_{\text{alive}}(G)$ within each generation $G$ that were still alive at the end of the observation period. The last generation $G_{\text{last}}$ to be included was then determined as the maximum generation such that

$$\frac{\sum_{G=1}^{G_{\mathrm{last}}} n_{\mathrm{alive}}(G)}{\sum_{G=1}^{\infty} n_{\mathrm{alive}}(G)} \leq 5\%. \tag{1}$$

All further generations were removed from the dataset. This procedure truncated the trees to 7, 6 and 5 generations, respectively for the three replicate experiments using our *MYCN* amplified cell line. *MYCN*-inhibited and rapamycin-treated trees were 5 generations deep.

### Spatial trend

To assess potential spatial biases related to locally variable conditions, cells were divided into a $4 \times 4$ grid according to their position at division. The cycle length distribution of cells within each grid region (containing $\geq 5$ cells) was compared to the distribution (1) within every other grid region and (2) of the whole dataset at a 5% significance level using a two-sided Kolmogorov-Smirnov test and correcting for multiple testing using the Benjamini-Hochberg procedure (using functions ks.test and p.adjust in R). Note also that because cells are motile they experience a range of local environments during their lifetime.

## Appendix 2

### Bifurcating autoregressive model for cycle inheritance

#### Setup

In order to explore systematically which simple local inheritance schemes can generate the experimentally observed cycle length correlations, we study a class of Gaussian latent variable models of adjustable complexity. In the models, the cycle length $\tau$ of a cell is obtained from a standard normal variable $\ell$ by a nonlinear transformation $\tau = g(\ell)$. In our data, cell cycle lengths are roughly log-normally distributed, so $g$ is approximately a shifted exponential function. To simplify and make the model more robust to outliers, we determine $g$ empirically, such that its inverse $g^{-1}$ transforms the cell cycle length into a standard normal variate. That is, we choose $g(\ell) = c_{\mathrm{ex}}^{-1}(c_{\mathrm{gauss}}(\ell))$ where $c_{\mathrm{gauss}}$ is the cumulative distribution function (CDF) of a standard normal distribution, and $c_{\mathrm{ex}}$ is the empirical CDF of the experimental data. This transformation discards all information about the shape and mean value of the cycle length distribution; the set of $\ell$ variables then purely reflect the strength of correlations between cell cycles. The (Pearson) moment correlation coefficients between the variables $\ell$ are identical to the so-called Gaussian rank correlation coefficients (*Boudt et al., 2010*) between the corresponding cycle lengths $\tau$, which are similarly robust to outliers as the more common Spearman rank correlation.

The Gaussian variable $\ell$ is used to model correlation by inheritance. $\ell$ is a weighted sum of $d$ latent, centered Gaussian variables $x = (x_1, \ldots, x_m)^T$ with positive weights $\alpha = (\alpha_1, \ldots \alpha_m)^T$, denoted as vectors $x$ and $\alpha$. That is, $\ell = \alpha^T x = \sum_l \alpha_l x_l$. Inheritance in the model occurs by passing on latent variables from mother to daughter cells. The basic model equation relation reads

$$x^i = \mathbf{A}x + b\xi^i + \bar{b}\xi^{\bar{i}}. \tag{2}$$

Here, a superscript $i = 1, 2$ denotes a daughter cell, and absence of a superscript refers to the mother cell. The matrix $\mathbf{A}$ implements inheritance: The average of a daughter's latent variables, given the mother's is $\langle x^i | x \rangle = \mathbf{A}x$. This linear coupling of latent variables through inheritance may take any form compatible with the basic stability requirement that its operator norm must satisfy $\|\mathbf{A}\| < 1$. Since both daughters inherit the same contribution from the mother, inheritance correlates the daughters' latent variables positively. Daughter cells are also subject to random fluctuations which we model by standard normal random vectors $\xi^i$. These fluctuations are correlated due to the term $\bar{b}\xi^{\bar{i}}$ in *Equation 2*. Here $\bar{i}$ designates the sister cell of $i$, for example $\bar{2} = 1$. We parametrize these correlations via

$$b = \cos(\beta/2), \; \bar{b} = \sin(\beta/2), \; \gamma \equiv 2b\bar{b} = \sin\beta, \text{where} -\pi/2 < \beta < \pi/2. \tag{3}$$

The sister correlations conditioned on the mother latent variables then become

$$\langle x^i x^{\bar{i}T} | x \rangle = \gamma \mathbf{I}; \quad \langle x^i x^{iT} | x \rangle = \mathbf{I}, \tag{4}$$

where $\mathbf{I}$ is the $d$-dimensional unit matrix. Positive sister correlations ($\gamma > 0$) may arise due to fluctuations that occur within the mother cell after its cycle duration has been fixed and are shared by the daughters. Negative correlations ($\gamma < 0$) may arise due to partitioning noise upon inheritance. Note that latent variable fluctuations are correlated between sisters but uncorrelated between different latent variables. Effectively, our choice of parametrization partitions all fluctuating cell cycle-relevant processes within the daughter cells into $d$ Gaussian components that are maximally decorrelated, similar to a principal component decomposition.

Overall, *Equation 2* defines an unbiased model with linear, local inheritance of latent variables, and an output that is a linear combination of latent variables. Its Gaussian form may be justified as the maximum-entropy distribution (*Jaynes, 1957*) for this problem, since only covariance information is used as an experimental input at this stage. Our model is a first-

order vector autoregressive process, defined on the lineage tree of cells; it is a latent-variable generalization of the bifurcating autoregressive model already considered by Staudte and coworkers (**Staudte et al., 1984**). We remark that in **Staudte et al. (1984)** the dimensionality is $d = 1$, and therefore $\alpha$ is unnecessary.

## Stationary exponential growth

Combining **Equations 2 and 4** by the law of total variance, the latent covariance satisfies

$$\langle x^i x^{iT} \rangle = \mathbf{A} \langle xx^T \rangle \mathbf{A}^T + \mathbf{I}. \tag{5}$$

In stationary exponential growth, averaging over a single lineage forward in time, a stationary distribution with mean $0$ and covariance $\mathbf{C}_\infty$ is established. Then $\langle x^i x^{iT} \rangle = \langle xx^T \rangle = \mathbf{C}_\infty$, and **Equation 5** implies

$$\mathbf{C}_\infty = \mathbf{A}\mathbf{C}_\infty \mathbf{A}^T + \mathbf{I} = \sum_{k=0}^{\infty} \mathbf{A}^k \mathbf{A}^{kT}. \tag{6}$$

We take this stationary distribution of latent variables as initial condition for root cells of lineage trees, assuming they come from an equilibrated growth phase. This assignment is not strictly correct because the stationary distribution along forward lineages is different from the distribution of all cells in an exponentially growing population at a given time (**Lin and Amir, 2017**); however, the difference was small for our parameters when tested numerically (not shown).

## Computation of the likelihood

We aim to compute the probability of generating a lineage tree with given cycle lengths within the model. We fix a minimum generation number and consider only trees in which essentially all branches reach this number, thus discarding overhanging cells on some branches, (see also **Sandler et al., 2015**). This is crucial since in experiments with finite duration, selection bias would otherwise be introduced (**Cowan and Staudte, 1986**).

We begin by indexing cells in a tree by their pedigrees, which are the sequences of sister indices counting from the root cell, for example $I = i_1 i_2 \ldots i_k$ for a cell in generation $k$ and $Ii_{k+1}$ for one of its daughters. Sorting these indices, we can then arrange all Gaussian-transformed cycle lengths in a tree into a single vector $\underline{\ell}$. Since $\underline{\ell}$ is Gaussian with mean $\underline{0}$, its log-probability takes on the simple quadratic form

$$\mathcal{P}(\underline{\ell}) = \mathcal{P}(\underline{\ell}|\mathbf{A}, \gamma, \alpha) = -\frac{1}{2}[\log \det(2\pi \underline{\mathbf{C}}_\ell) + \underline{\ell}^T \underline{\mathbf{C}}_\ell^{-1} \underline{\ell}]. \tag{7}$$

To evaluate **Equation 7**, we need to determine the joint covariance matrix $\underline{\mathbf{C}}_\ell$ of Gaussian cycle lengths over the given tree structure as a function of the parameters $\mathbf{A}, \gamma, \alpha$. We start by first deriving the joint covariance matrix $\underline{\mathbf{C}}$ of the latent variables $\underline{x}$. This is a block matrix with $d \times d$ blocks $\mathbf{C}^{IJ}$ that correspond to pairs of cells in well-defined relationships, such as mother-daughter, cousin-cousin, etc. Since the lineage tree is sampled from stationary growth, $\mathbf{C}^{IJ}$ depends only on the relationship of $I$ and $J$, that is on their respective ancestral lines up to the latest common ancestor, and not on the history before. In particular, if $I = Ji_{k+1} \ldots$ then cell $I$ is a descendant of cell $J$ and we write this as $I \succ J$; otherwise we write $I \nsucc J$. Note that $I \nsucc I$.

Splitting one cell pedigree as $I = Ki$, from **Equation 2** we derive the relations

$$\langle x^{Ki} x^{JT} \rangle = \langle (\mathbf{A} x^K + b\xi^i + \bar{b}\xi^{\bar{i}}) x^{JT} \rangle = \begin{cases} \mathbf{A} \langle x^K x^{JT} \rangle & J \nsucc K \quad (i) \\ \mathbf{A} \langle x^K x^{KT} \rangle \mathbf{A}^T + \gamma \mathbf{I} & J = K\bar{i} \quad (ii) \end{cases} \tag{8}$$

**Equation 8** lets us compute $\mathbf{C}^{IJ}$ by a recursive procedure, as follows:

- Consider the case $I \succ J$. If $I = Ki$, then $J \nsucc K$. Now use **Equation 8 (i)** repeatedly ($k$ times), moving up the ancestral line, until arriving at the form $\mathbf{C}^{IJ} = \mathbf{A}^k \langle x^J x^{JT} \rangle = \mathbf{A}^k \mathbf{C}_\infty$.

- In the case $I = Ki_1 \ldots i_k$, $J = Kj_1 \ldots j_{k'}$ with $j_1 = \bar{i}_1$, no cell is a descendant of the other, and their last common ancestor is $K$. Use **Equation 8 (i)** (or its transpose) on both branches repeatedly until the form $\mathbf{A}^{k-1} \langle \boldsymbol{x}^{Ki_1} \boldsymbol{x}^{K\bar{i}_1 T} \rangle \mathbf{A}^{k'-1 T}$ is obtained. Then use **Equation 8 (ii)** to get $\mathbf{C}^{IJ} = \mathbf{A}^k \mathbf{C}_\infty \mathbf{A}^{k' T} + \gamma \mathbf{A}^{k-1} \mathbf{A}^{k'-1 T}$.

These two cases cover all possible cell-cell relations, so that the procedure fully determines the joint latent covariance $\underline{\mathbf{C}}$ for a given tree structure, inheritance matrix $\mathbf{A}$ and sister correlation $\gamma$.

Finally, to obtain the covariance $\underline{\mathbf{C}}_\ell$ of the Gaussian cycle lengths $\ell$, we project onto $\alpha$. The elements of $\underline{\mathbf{C}}_\ell$ result as

$$C_\ell^{IJ} = \langle \ell^I \ell^J \rangle = \alpha^T \langle \boldsymbol{x}^I \boldsymbol{x}^{J T} \rangle \alpha = \alpha^T \mathbf{C}^{IJ} \alpha. \tag{9}$$

This completes the evaluation of the log-probability $\mathcal{P}$ (**Equation 7**), which is also equal to the log-likelihood of the model, $\mathcal{P}(\underline{\ell}) = \mathcal{L}(\mathbf{A}, \gamma, \alpha)$. Accounting for the constraint $\langle \ell^2 \rangle = 1$ which we impose to fix the arbitrary normalization of $\ell$, the full model has $d^2 + 1 + d - 1 = d(d+1)$ adjustable parameters. This number can be reduced by restricting the inheritance matrix to a specific form, or by setting $\gamma = 0$, as was done for the model variants discussed in the main text.

As a corollary, the Gaussian rank correlation between cycle lengths of any pair of cells results as

$$\rho_{\tau^I \tau^J}^{\text{gauss}} = \rho_{\ell^I \ell^J} = \frac{C_\ell^{IJ}}{C_\ell^{II}} = \frac{\alpha^T \mathbf{C}^{IJ} \alpha}{\alpha^T \mathbf{C}_\infty \alpha}. \tag{10}$$

In the one-dimensional special case $d = 1$, the projection on $\alpha$ becomes irrelevant and **Equation 10** reduces to

$$\rho_{\tau^I \tau^J}^{\text{gauss}} = \begin{cases} a^k & I > J \text{ or } J > I \\ a^{k+k'} + \gamma a^{k+k'-2}/c_\infty = a^{k+k'-2}[a^2 + \gamma(1-a^2)] & I \not> J \text{ and } J \not> I \end{cases}, \tag{11}$$

where $k, k' \geq 1$ the distances to the latest common ancestor as in the algorithm above, and $a \equiv \mathbf{A}$ is the $1 \times 1$ inheritance matrix. Some special cases of **Equation 11** given already in **Cowan and Staudte (1986)** are $\rho_{\text{ss}}^{\text{gauss}} = [a^2 + \gamma(1-a^2)]$ for sisters and $\rho_{\text{c1}}^{\text{gauss}} = a^2 \rho_{\text{ss}}^{\text{gauss}} = \rho_{\text{md}}^{\text{gauss} 2} \rho_{\text{ss}}^{\text{gauss}}$ for first cousins. In other words, to compute the correlation between related cells, one multiplies mother-daughter correlations along the path connecting them, taking a shortcut via the daughters of the last common ancestor where one instead multiplies with the sister-sister correlation. Specializing further to $\gamma = 0$, **Equation 11** reduces to the well-known relation $\rho_{\tau^I \tau^J}^{\text{gauss}} = \rho_{\text{md}}^{\text{gauss} k+k'}$ where $k + k'$ is the number of cell divisions linking $I$ and $J$. As detailed in the main text, these one-dimensional special cases are insufficient to explain our data.

## Evidence calculation

To compare different model versions' ability to explain but not overfit the data, we employed a standard Bayesian model selection scheme (see e.g. **Wasserman, 2000**; **MacKay, 2003**). Within this framework, model selection is treated on the same grounds as parameter inference; the task is to assign to each one out of a set $\mathcal{M}$ of models its likelihood to have generated the data. One or several plausible models can then be selected on these grounds. Concretely, the scheme proceeds as follows. The probability of model $M$ to generate data $\underline{\ell}$ is obtained by integrating over all parameter values $\pi_M$, which are distributed over a parameter space $\Pi_M$ with prior distribution $p(\pi_M | M)$:

$$p(\underline{\ell} | M) = \int_{\Pi_M} p(\underline{\ell} | \pi_M, M) p(\pi_m | M) d\pi_M. \tag{12}$$

Here, $p(\underline{\ell} | \pi_M, M) = \exp[\mathcal{P}(\underline{\ell})] = \exp[\mathcal{L}(\pi_M)]$ is equal to the likelihood function for model $M$,

**Equation 7.** Applying Bayes' rule, the probability that among $\mathcal{M}$, $M$ was the model that generated the data, is then

$$p(M|\underline{\ell}) = \frac{p(M)p(\underline{\ell}|M)}{\sum_{\mathcal{M}} p(M')p(\underline{\ell}|M')}. \tag{13}$$

When models are equivalent a priori as we assume here, then both the prior belief in a model, $p(M)$, and the entire denominator in **Equation 13** are unimportant constants. Then the so-called evidence or Bayes factor, obtained by calculating

$$E(M) = \int_{\Pi_M} p(\underline{\ell}|\pi_M, M)\, d\pi_M, \tag{14}$$

is proportional to each model's probability of having generated the data, **Equation 13**. We calculated $E(M)$ numerically by Monte-Carlo integration of **Equation 14**; in the main text we show the evidences relative to Model V. Conventionally, an advantage in $E$ of a factor of 10 or more is considered strong support in favor of a model.

We briefly discuss some important features of model selection by evidence. In the asymptotic case of large samples (not applicable for the present data), the evidence $E$ is approximated by the well-known Bayes information criterion (BIC), which is an alternative to the popular Akaike information criterion (AIC). While AIC is constructed to select a model whose predictions are maximally similar to future repetitions of the same experiment, evidence and BIC select the model that is most likely to have generated the existing data. BIC and evidence, but not AIC, have a desirable consistency property: If the models $\mathcal{M}$ are recruited from a hierarchy of nested models which also contains the true model, then the simplest model in $\mathcal{M}$ comprising the true model is always favored for large enough samples (**Wasserman, 2000**). This consistency is a manifestation of a general preference of the evidence for parsimonious models. To illustrate this point, following **MacKay (2003)**, we expand the log evidence around the maximum a-posteriori estimate $\pi_M^*$, using Laplace's method:

$$\log E(M) \simeq \mathcal{L}(\pi_M^*) + \log\det\left(2\pi H_{M*}^{-1}\right) - \log\mathrm{vol}(\Pi_M) \simeq \mathcal{L}(\pi_M^*) + \sum_{i=1}^{d} \log\frac{s^i}{S^i}. \tag{15}$$

Here, $H_{M*} = \left(\frac{\partial^2 \mathcal{L}}{\partial_{\pi_M^i}\partial_{\pi_M^j}}\right)\big|_{\pi_M^*}$ is the Hessian of the log-likelihood, and for simplicity we have assumed a flat parameter prior $p(\pi_M) = 1/\mathrm{vol}(\Pi_M)$. The ith eigenvalue $2\pi/s_i$ of $H_{M*}$ determines the width $s_i$ of the peak of the posterior distribution around $\pi_M^*$, along the ith principal axis. In the last equality, we have written the parameter space volume $\mathrm{vol}(\Pi_M) = \prod_i S_i$ as a product of parameter ranges $S_i$ along the principal axes. **Equation 15** can be interpreted as follows: As more parameters are added to a model, the fit accuracy, measured by $\mathcal{L}(\pi_M^*)$, generally improves. However, each new parameter $i'$ incurs a penalty $\log(s_{i'}/S_{i'}) < 0$. The more the new parameter needs to be constrained by the data, the more the evidence is reduced. Thus the basic mechanism of parsimony in Bayesian model selection is this: Complex models are characterized by a large number of parameters with wide a priori allowed ranges and sensitive dependence on the data; in other words, they require the data to pick parameters from a large set of possibilities. Complex models are penalized and ranked as less likely. Indeed, such an overly flexible model can be fitted to diverse data, which we should expect to diminish the support that a particular set of data can give to it.

## Numerical efficiency and implementation

The evaluation of $\mathcal{L}$ (**Equation 7**) requires a final numerical inversion of the recursively assembled covariance $\underline{\mathbf{C}}_\ell$ to obtain the stiffness matrix, which costs $O(N^3)$ operations, where $N$ is the number of cells in the tree. To reduce this cost, it is possible to devise an equivalent recursive scheme in which the projected stiffness matrix, not the covariance matrix, is computed recursively, using efficient block-inverse formulæ in each recursive step. The

computational complexity can then be improved to $O(N^2)$. A version of this alternative, equivalent scheme was implemented in the programming language OCaml and used to obtain the results presented in the main text.

## Appendix 3

# The growth-progression model

## Setup

The growth-progression model is based on the idea that the two cell-cycle controlling processes are cell-cycle progression and cell growth. The state variables of both processes, namely, the timing of the regulatory license to divide encoded by $p$, and cell size $s$, respectively, are inherited from mother to daughter. The two processes are coupled via inheritance as detailed in the following.

## The cell-cycle progression process

Inheritance of the velocity of cell-cycle progression is modeled by a fluctuating, centered Gaussian variable $q$, passed on from mother to daughter, entirely analogous to version II of the BAR model (see there for additional explanation), according to

$$q_i = aq + b\xi_i + \bar{b}\xi_{\bar{i}}, \tag{16}$$

where subscript $i = 1,2$ denotes a daughter cell, $\bar{i}$ its sister, and no subscript, the mother. $a$ with $|a| < 1$, implements inheritance. The intrinsic fluctuation strength and coupling is given by $b$ and $\bar{b}$; effectively, daughter cells are correlated for given mother by, $\langle q_i q_{\bar{i}} | q \rangle = \gamma$; $\langle q_i^2 | q \rangle = 1$, where $\gamma = 2b\bar{b}$. From the centered $q$ variables, shifted and scaled Gaussian variables $p = \sigma_p q + \mu$ were generated, finally yielding log-normal regulation cycle durations $\tau_p = \exp(p)$. The duration $\tau_p$ is the time elapsed since the last division until regulatory license is given to divide again. Overall, the progression process has four adjustable parameters, $\mu$, $\sigma_p$, $a$ and $\gamma$.

## The growth process

The growth duration $\tau_g$ is defined as the time to grow from an initial size $s_b$ to the threshold size $s_{th}$. Size accumulation was modeled by exponential growth with the exponential growth rate

$$\frac{ds}{dt} = ks. \tag{17}$$

However, under *MYCN* inhibition, exponential growth was prone to generate unreasonably large cells. Here we instead modeled growth by the logistic growth process

$$\frac{ds}{dt} = k\left(1 - \frac{s}{s_{max}}\right)s, \tag{18}$$

where $k$ is the growth rate constant and $s_{max}$ the maximum cell size. We fixed $s_{max} = 20$, to match the approximate cell size at which the growth rate starts to decrease with observations (*Sung et al., 2013*; *Tzur et al., 2009*). The two growth laws *Equations 17, 18* yield

$$\tau_g = k^{-1} \log\left(\frac{s_{th}}{s_b}\right) \tag{19}$$

and

$$\tau_g = k^{-1} \log\left[\frac{s_{th}(s_b - s_{max})}{s_b(s_{th} - s_{max})}\right], \tag{20}$$

respectively.

The normalized threshold cell size $s_{th}$ fluctuates slightly and independently in each cell as $s_{th} = 1 + \zeta$ with $\zeta \sim \mathcal{N}(0, \sigma_g^2)$. At division, the final mother cell size $s_{div}$ is halved, with each

daughter receiving a new size at birth $s_b = s_{div}/2$. Effectively, for the subset of cell cycles that are limited by growth, this process corresponds to a sizer mechanism *Facchetti et al. (2017)*. The growth process has two adjustable parameters, $k$ and $\sigma_g$.

### Coupling progression and growth

The two processes are coupled via a checkpoint which requires both to be completed before a cell has license to divide. The cell-cycle length is then determined as $\tau = \max(\tau_p, \tau_g)$. If division is stalled by insufficient cycle progression, $s$ continues to accumulate until cell division, so that the final cell size $s_{div} > s_{th}$. Thus, inheritance of the growth process is influenced by the cycle progression process. In contrast, the progression process is inherited in an autonomous fashion. This unidirectional inheritance structure recapitulates the unidirectional coupling between hidden processes found in the preferred BAR models IV and V.

### Model simulation

The growth-progression model was implemented in Matlab (R2016b), R (3.4.3) and OCaml (4.06) (with identical results but increasing execution speed) and lineage trees were simulated. For each tree, an initial cell was generated with birth size $s_b = s_{th}/2$ and $p = \mu$ and subsequently, an unbranched single lineage was simulated for 100 generations for equilibration. Its final cell was used as founder cell for the tree. For the data shown, 30 trees of 7 generations each were simulated, roughly corresponding to the dataset sizes obtained experimentally. The simulation was repeated 100 times to generate confidence bounds.

### Parameter optimization

Parameters were fitted using Approximate Bayesian Computation independently for each dataset. In an adaptive procedure, (sometimes non-uniform) prior distributions of model parameters were first generated. For each parameter set, 500 trees were simulated at a depth of 7 generations. To compare data $D$ and simulations $\hat{D}$, we used a set $S$ of summary statistics composed of the nine correlation coefficients as shown in *Figure 3C*, and the mean and all quartiles of the distribution of cell-cycle durations. A squared-distance between data and simulation summary statistics was calculated as

$$\rho^2[S(\hat{D}), S(D)] = \sum_{i=1}^{N} \left[ \frac{S(D_i) - S(\hat{D}_i)}{\sigma_i} \right]^2, \tag{21}$$

where $\sigma_i$ were calculated based on the data bootstrap confidence (95%) bounds. Samples of the approximate posterior probability distribution of parameter values (Figure 3—figure supplement 3A ) were then generated by accepting parameter sets with $\rho^2(S(\hat{D}), S(D)) < \epsilon$, weighted by the inverse of the local prior density in parameter space. We used a tolerance $\epsilon = 2$ but results were not sensitive to the choice of $\epsilon$ within the range $1 \ldots 4$. Prior parameter ranges were extended as far as necessary for accepted sets to converge. The final accepted ranges of each parameter was used as credible regions, and the medians of each parameter distribution from these samples were selected as best-fits for further analysis.

