## [Decision Letter]

**Acceptance summary:**

Kuchen et al. have developed a general model that can explain the high degree of intra-generational correlation in cell cycle times that have been observed in multiple systems. Based on insight provided by this general model, they show that these correlations can arise from a unidirectional coupling of cell size and cell-cycle speed by a minimal cell-size checkpoint, which they verify using perturbation experiments.

**Decision letter after peer review:**

Thank you for submitting your article "Hidden long-range memories of growth and cycle speed correlate cell cycles in lineage trees" for consideration by *eLife*. Your article has been reviewed by three peer reviewers, and the evaluation has been overseen by Naama Barkai as the Senior and Reviewing Editor. The reviewers have opted to remain anonymous.

The reviewers have discussed the reviews with one another and the Reviewing Editor has drafted this decision to help you prepare a revised submission.

Summary:

Kuchen et al. have developed an interesting and general model that can explain the high degree of intra-generational correlation in cell cycle times that have been observed in multiple systems. First, using Bayesian inference on their own time-lapse data, they find that a model of two unidirectionally coupled heritable components that affect cell-cycle time best explains their data. They go on to propose that these components can be cell size and cell-cycle speed, provided that growth and division are coupled via a minimum size checkpoint. They then verify this model through perturbation experiments.

Essential Revisions:

Please address all comments below.

In particular please pay a special attention to the following two points:

1) The concerns over reproducibility between datasets,

2) Provide much clearer explanation of the unbiased model-selection.

Reviewer #1:

The paper studies division time correlations within neuroblastoma cells through time-lapse microscopy. Their approach uses unbiased statistical models to infer a minimal model that they claim recapitulates the experimentally observed correlations within populations of cells based on a set of heritable factors influencing interdivision times. The authors then study a model of division times based on growth to a threshold size and the inheritance of cell cycle progression factors that determine the rate of passage through the cell cycle. The authors claim that this second model recapitulates WT data in addition to matching the correlations observed in cells either grown with rapamycin (to limit cell growth) or cells with a reduced rate of passage through the cell cycle obtained by restricting the expression of the oncogene MYCN.

I found their unbiased statistical approach of particular interest since they considered a range of hierarchical models and used a metric for evaluating them which accounts for potential effects of overfitting with larger numbers of fitting parameters. However, to interpret this section with a critical eye requires more intuition for the metric of discrimination between these hierarchical models. In particular, Equation 12 which appears to address this should be referenced and discussed in the main text. With regards to the rest of the paper, the predictions of the authors when varying the limitations of either growth or cell cycle progression were unclear. This would be greatly aided by giving the reader intuition about what one would expect for either "pure" model (either a pure sizer for size control or cells which are completely limited by cell cycle progression).

Additionally, the following main comments should be addressed:

1) How are the fitting parameters re-evaluated when inhibiting growth or cell cycle progression relative to the control case?

2) Why do the authors switch to logistic growth during the MYCN growth? This is currently not even mentioned in the main text. It is concerning that in order to fit the MYCN growth case this fundamental assumption needed to be altered.

3) Is the agreement between the model predictions and the observed division time correlations robust to variations in the cell size control strategies assumed? There is increasing evidence that many mammalian cells follow an "adder" model (see for example Cadart et al., 2018 which is incorrectly referred to in the paper as supporting a critical size), so testing the model for this size control strategy would be an important test of the proposed model.

4) The choice of cell cycle timing as a feature of interest was not adequately justified. The results would be stronger if the authors can use data on correlations in cell size between relatives to test Model V and its predictions.

5) The explanation of the results shown in Figure 4 does not provide clear intuition for why the correlations should be altered in the observed fashion. It would be helpful if the authors took more care when explaining this. For example: The authors show in Figure 3E that growth-limited cells have negative mother-daughter correlations in cell cycle timing. This being the case, why is it that the mother-daughter correlations in the presence of rapamycin (Figure 4D) are in fact larger than those correlations measured in the MYCN treated cells?

Reviewer #2:

In the present manuscript, Kuchen and colleagues investigate how regulation of cell cycle progression and cell growth are coupled using analysis of extended cell lineages and mathematical modeling. Initially, they performed time-resolved live-cell microscopy of a MYCN – over-expressing neuroblastoma line to generate cell lineage trees spanning 5-7 generations. As previously reported, they observed higher intra-generational than ancestral correlations, which has so far been attributed to coupling between cell cycle and circadian clock (Sandler et al., 2015, Chakrabarti et al., 2018). The authors now took an unbiased approached based on a Gaussian latent variables and propose a model where two unidirectionally coupled variables are inherited to daughter cells. They mechanistically implement these two variables as cell growth and cell-cycle progression and show that it is sufficient to couple both through a minimal-size checkpoint to account for the observed correlation patterns in lineage trees. This model is supported by experimental perturbations of cell growth (via mTOR inhibition) and cell cycle progression (decreasing MYCN levels) as well as by analysis of existing lineage data from rapidly cycling human ES cells.

The paper is well written and easy to follow. The authors present a very plausible alternative explanations for the non-intuitive correlation patterns in lineage trees and provide further evidence for a size checkpoint in mammalian cell cycle regulation. As the focus of the manuscript is on the use of abstracted mathematical models to explain a complex cellular phenotype (inheritance of cell cycle length), there is little mechanistic/molecular inside regarding the details of the observed phenomena (e.g. minimal cell size checkpoint). However, the provided experimental validation provides strong support for the presented hypothesis. As an experimentalist, I would have liked to see additional orthogonal perturbations to further strengthen the data, but believe that this would be beyond the scope of this study.

Main points:

1) While the modelling results match the data within the estimated error bounds, the correlation patterns seem to be less pronounced. For example, the increased correlation to first cousins compared to aunts in Figure 3C (and Figure 3—figure supplement 1D) is hardly reproduced in the model. Similarly, the correlations to the removed side-branch seem to be about equal for all generations in the model, while the data indicates increased intra-generational correlations. It would be helpful if the authors comment on and explain such deviations or limitations of the model.

2) As a main assumption of the authors is that MYCN over-expression disrupts the circadian clock which has been previously linked to the inheritance patterns of cell cycle length, they should validate this in the cell line used. This could be done for example by quantifying BMAL1 expression in the presence/absence of doxycycline (see Figure 4—figure supplement 1A.

Reviewer #3:

Kuchen et al. have developed an interesting and general model that can explain the high degree of intra-generational correlation in cell cycle times that have been observed in multiple systems. First, using Bayesian inference on their own time-lapse data, they find that a model of two unidirectionally coupled heritable components that affect cell-cycle time best explains their data. They go on to propose that these components can be cell size and cell-cycle speed, provided that growth and division are coupled via a minimum size checkpoint. They then verify this model through perturbation experiments. Although the paper appears to be of general interest to *eLife* readers, there are several points that require clarification before publication.

1) Representation of data. The authors present 3 repeats of the time-lapse microscopy for neuroblastoma TET21N cell lineages. (Figure 1E). For the Bayesian analysis (Figure 2) they only show the different models plotted against Repeat 1 in the main and supplemental figures (which appears to be the bottom dataset of Figure 1E – it would be helpful if the different repeats were labelled in this figure). However, there appears to be significant differences between the 3 repeats shown in Figure 1E. Although it is difficult to see (in part because the horizontal line across each axis is at a different value of the Cross correlation coefficient – how was this value selected?) it appears that in repeats 2 and 3 the level of correlation in distant side branches is much lower (around 0). This appears particularly worrying given the results of the Bayesian analysis for Model VII (Figure 2—figure supplement 1). Although this model results in no correlation past first cousins, it gives the best fit to data for repeat 2 and 3 (better in fact than model 5, Figure 2—figure supplement 1), and only fails to reproduce replicate 1. This suggests that for 2 out of 3 datasets a model that assumes no correlations past first cousins is being selected as the best model from the Bayesian analysis. The authors should address this issue, and show plots of the Bayesian analysis against all 3 repeats.

In the rest of the figures, which dataset chosen is not consistent. Although they do display a fit of their growth/progression model to repeat 1 and 2 in the Figure 3—figure supplement 1, they now chose repeat 3 to display in Figure 3 – which appears to be the best fit to the model. In Figure 4, panel C shows cell-cycle lengths for repeat 3, but it is not clear which repeat(s) is/are used in the calculation of the correlation ratios. It would be helpful if the authors were consistent with their data selection in the main figures and supplements throughout the paper.

2) Experimental evidence for cell size/cell cycle checkpoint model. The authors relate latent variables x_1_ and x_2_ to cell size and the cycling of cell cycle factors respectively. However, since these quantities are not dynamically tracked throughout cell growth, I have some questions about the authors' interpretation and model. First, the model assumes the existence of a size threshold. If size does not affect cell cycle progression (another assumption), that suggests small-born cells are under a sizer-type control, whereas large-born cells (progression limited) are under a timer-type control. Such a model seems difficult to square with ongoing work in other organisms, which typically find adder-type (many single-cell microbes but also a number of mammalian cell lines, cf. Cadart et al., 2018) or sizer-type (e.g. fission yeast) control. Could the authors comment on how their model is biologically justified and how does it relate to alternative models of cell growth? Timer control is usually not considered because, on its own, it does not generate stable cell size. Indeed, in the appendix the authors note that, in the progression-limited regime, their model generates cells that are too large, so they implemented logistic, rather than exponential growth. In my understanding, this effectively implements a sizer again. What evidence do the authors have for these assumptions?

On the other hand, in subsection “Effects ofmolecular perturbations on cell-cycle correlations correctly predicted by the model” the authors claim that the growth of MYCN-inhibited cells is not affected. How did the authors verify this? If growth is logistic, as in the model, then growth rate is clearly affected beyond a certain size. Naively, I would also expect slower growth rates if growth is exponential and cell size is homeostatic. Additionally, the effects on cell size due to the addition of rapamycin appears weak (Figure 4B) – Is it possible to quantify this difference, with an estimation of errors?

---

## [Author Response]

Reviewer #1:[…]However, to interpret this section with a critical eye requires more intuition for the metric of discrimination between these hierarchical models. In particular, Equation 12 which appears to address this should be referenced and discussed in the main text.

The reviewer asks us to elaborate on model selection. We agree that this is a critical issue. In response, we have added in the main text a short intuitive explanation on Bayesian evidence used for model selection: “Selection is based on the Bayesian evidence, which rewards fit quality while naturally penalizing models of higher complexity (defined as being able to fit more diverse data sets; for details see Appendix 2, “Evidence calculation”).” To underpin this, we have greatly expanded on the fundamentals of Bayesian evidence and how we calculated it in Appendix 2, section Evidence calculation. Particular emphasis is placed on an intuitive explanation of how the penalty for overfitting is built-in to Bayesian model selection. While rather intuitive, this detailed explanation requires some technical preliminaries and recapitulates prior work of others. We therefore present it in an Appendix. In addition to the book by MacKay previously cited, we now also cite a review by Wasserman, 2000, that gives a lucid methodological overview of this topic.

With regards to the rest of the paper, the predictions of the authors when varying the limitations of either growth or cell cycle progression were unclear. This would be greatly aided by giving the reader intuition about what one would expect for either "pure" model (either a pure sizer for size control or cells which are completely limited by cell cycle progression).

The reviewer points out that developing intuition on how the growth-progression model functions would help understand the outcome of the perturbation experiments. We thank them for making us think further in this direction. We have now made several additions throughout the paper that build upon each other in giving an intuitive view on how cell growth and cell-cycle progression interact in our model to yield the observed correlation patterns of cell-cycle length. Before outlining this new material, we first remark that a pure “sizer” model of the type proposed here is a degenerate case that may obscure, rather than enlighten, what happens in our model. A pure sizer would simply cause exactly equal (no threshold noise) or slightly anticorrelated (with noise) cell-cycle lengths in mother and daughters and have no long-range memory whatsoever. Of note, even the very fast proliferating ESCs in our manuscript are not described by a pure sizer. More generally, a pure sizer is also an unrealistic scenario in the light of recent data (e.g., Cadart et al., 2018, showing near-adder, rather than sizer, behavior for many mammalian cell lines), and hence we chose not to discuss it, mainly to avoid confusion. In the new sections building intuition for the model behavior, we emphasize from the start how growth and cycle-progression interact in generating long-range cell-cycle correlations.

This intuitive explanation: “While growth and cell-cycle progression are separable and heritable processes, they also interact. […] Thus, despite inheritance of growth and cell-cycle regulators mothers and daughters may have very different cycle lengths due to this interaction.” This interaction is then the basis for both rapidly decaying ancestral (e.g., mother-daughter) and long-range intragenerational correlations in cell-cycle length. This is now explained in a new paragraph: “To gain intuition …” The new paragraph makes use of Figure 3E (already present previously), showing that mother and daughter cell cycles become anticorrelated due to the above-described influence of cell-cycle progression on growth.

To clarify the origin of long-range intragenerational correlations further, we have now added a new Figure 3F (complemented by new Figure 3—figure supplement 1F) and a new explanation: “While cousins are positively correlated overall, this correlation is carried specifically by cousins that descend from a grandmother with a progression-limited cell cycle (Figure 3F, blue dots), whereas cousins stemming from a growth-limited grandmother are hardly correlated (Figure 3F, orange dots). Since progression-limited cells can grow large, this observation indicates that cousin correlations are mediated by inheritance of excess size, as is confirmed by conditioning cousin correlations on grandmother size…”

These findings show that both low mother-daughter correlations and high intragenerational (e.g., cousin) correlations are *emergent* phenomena of the interplay between growth and cell-cycle progression. This intuition is then further developed in an amended paragraph on the perturbation experiments. In that paragraph, we found it quite helpful to first discuss the limiting case of a pure “timer”, i.e. pure progression-limited cycles, as this is actually approximated experimentally by MYCN inhibition. Hence we have taken up, at least in part, the above concrete recommendation by the reviewer.

Additionally, the following main comments should be addressed:1) How are the fitting parameters re-evaluated when inhibiting growth or cell cycle progression relative to the control case?

This question relates to two points: (i) prediction of the effects of perturbations and (ii) application of the model to the experimental data from the perturbation experiments. First, for the prediction of perturbation effects (Figure 4A), we used the parameters for TET21N cells control conditions (replicate rep1). To simulate inhibition of cell growth, we lowered the growth rate *k* by 10%. To simulate inhibition of cell-cycle progression, we increased the average cycle length via increasing the mean log of the progression time μ by 7.5%. These details were inadvertently omitted from the Materials and methods for which we apologize; we have now added these. Moreover, for ease of reference, we have now added a table with the numerical values of the parameters used in all the simulations as the new Figure 3—figure supplement 2, which complements Figure 3—figure supplement 1A, graphically depicting these values and their confidence bounds.

Second, after performing the perturbation experiments for growth and cell-cycle progression (using rapamycin and doxocyclin-inducible downregulation of MYCN, respectively) we refitted the model to the experimental data, subjecting all model parameters to optimization. For inhibition of cell-cycle progression, we found that the only parameter that showed a consistent difference between these perturbation data and control replicates 1-3 was μ, which was larger, corresponding to a longer cell-cycle duration (see Figure 3—figure supplement 1A, -myc1 and -myc2). For inhibition of cell growth, several parameters changed in an informative manner. Consistent with the known rapamycin effect, the estimated growth rates were either at the lower end (rap2) or strongly decreased (rap1) by comparison with the values for the control experiments. At the same time, mother-daughter and sister cell-cycle length correlations (*a* and g, respectively) increased. Mechanistically, such a tendency would be caused, for instance, by larger depletion of cell-cycle inhibitors. This finding is consistent with the action of rapamycin, which inhibits mTOR and hence growth, but leaves ERK and PI3K activities unaffected, which drive cell-cycle progression (e.g., Adlung et al., 2017). Hence, slower growth would delay attainment of the size threshold and thereby allow longer phases of degradation of inhibitors such as p21, which in turn should increase mother-daughter and sister correlations (Spencer et al., 2013). We added these results and a brief discussion of their biological significance in subsection “Cousin correlations reflect active cell-size checkpoint”, in connection with Figure 4.

2) Why do the authors switch to logistic growth during the MYCN growth? This is currently not even mentioned in the main text. It is concerning that in order to fit the MYCN growth case this fundamental assumption needed to be altered.

The reviewer correctly observes that when modeling cells with low MYCN, which proliferate slowly, we implement a limitation of growth rate via logistic growth, whereas in all other cases, we do not. This was done in this manner because we aimed at the simplest possible version of the growth-progression model for each particular situation. For all cases but the slowly growing and large low-MYCN cells, we obtained well-bounded distributions of the “size” (or resource) variable with a constant accumulation rate, i.e. exponential growth. To achieve this for low-MYCN cells, we needed to implement negative regulation of accumulation of the size variable, by introducing a logistic dependence of accumulation rate on size. This regulation kicks in at high values of the size variable that are very rarely reached when cells proliferate more rapidly, i.e. in high-MYC cells. Hence some form of growth limitation will surely exist, but it is largely inconsequential for the more rapid cell cycles. We have also verified that including growth limitation has only minor effects on the model fit results in the control and growth-inhibited conditions. In detail, we have made the following changes in response to this point of the reviewer:

1) We explain more clearly in the main text that logistic growth was used for low-MYCN cells (first when introducing the model and then again in connection with fits to perturbation experiments, Figure 4 C, D).

2) We now show in addition that the incorporation of a logistic growth limitation does not significantly alter the best-fit parameters and the corresponding fitted cell-cycle correlations in lineage trees when cell cycles are faster, as in TET21N control conditions and under rapamycin treatment. These results are shown in the new Figure 4—figure supplement 2. Hence leaving out growth limitation for faster growing cells is a valid approximation here.

3) We now give a more concrete perspective on the implications of our findings on inheritance of cell-cycle speed and cell size for cell-size regulation, (Discussion paragraph two), see also our response to point 3) below.

3) Is the agreement between the model predictions and the observed division time correlations robust to variations in the cell size control strategies assumed? There is increasing evidence that many mammalian cells follow an "adder" model (see for example Cadart et al., 2018 which is incorrectly referred to in the paper as supporting a critical size), so testing the model for this size control strategy would be an important test of the proposed model.

The reviewer raises the important question of size control strategies used by cells and refers to one of the basic phenomenological models, the adder model (a similar question was posed by reviewer #3, Point 2).

We thank the reviewer for pointing out the ambiguous referencing of Cadart et al., 2018, which we have now corrected and expanded as follows: “…shown that growth of many mammalian cells during the cell cycle adds a volume that only weakly increases with cell volume at birth (termed near-adder behavior”. Cadart et al. show that different mammalian cells fall within a continuum, ranging from sizer over adder to timer behavior, when quantifying cell size gain within one cell cycle. Our manuscript focuses on inheritance mechanisms that explain correlation patterns between cycles of related cells. Interestingly, our findings touch upon size control strategies, but from a different angle than the phenomenological timer, adder and sizer models. Our growth-progression model in effect combines “timer” cycles, when cycle duration is controlled by cell-cycle progression, and “sizer” cycles, when cycle duration is controlled by growth. Overall, the mix leads to a pattern of mitotic size versus birth size that lies between a pure timer and a pure adder, a behavior that has been observed experimentally by Cadart et al. for mammalian cell lines (Author response image 1). This finding indicates that our model is principally consistent with findings by Cadart et al., although it does not explicitly implement an adder mechanism. Indeed, adder and sizer are phenotypic descriptions of cell growth behavior that could be realized by different types of cellular mechanisms. Given that the mechanistic basis of the fundamental findings by Cadart et al. is not fully understood, we regard it as beyond the scope of the current manuscript to test various hypotheses with our model. We indicate now in closing the Discussion that this may be a fruitful direction for future work.

**Author response image 1. respfig1:** Simulation of the growth-progression model (with best-fit parameters for control rep1). The linear regression of final cell size versus cell size at birth shows a slope between 1 (pure adder) and 2 (pure timer), falling in the range of experimental data measured for mammalian cell lines by Cadart et al., 2018.

4) The choice of cell cycle timing as a feature of interest was not adequately justified. The results would be stronger if the authors can use data on correlations in cell size between relatives to test Model V and its predictions.

As explained in our response to Point 3, our main focus is on the mechanisms that determine cell-cycle timing.

5) The explanation of the results shown in Figure 4 does not provide clear intuition for why the correlations should be altered in the observed fashion. It would be helpful if the authors took more care when explaining this. For example: The authors show in Figure 3E that growth-limited cells have negative mother-daughter correlations in cell cycle timing. This being the case, why is it that the mother-daughter correlations in the presence of rapamycin (Figure 4D) are in fact larger than those correlations measured in the MYCN treated cells?

We agree that an intuitive explanation of the observed changes in correlation patterns was lacking. To give such an intuition, we need to interrogate the predictions of our growth-progression model. We have tried to do so in several places. First, we have added a more extensive discussion of how the interplay of growth and progression produce mother-daughter and cousin-cousin correlation (in subsection “Cell size and speed of cell-cycle progression are antagonistic heritable variables”, connected to Figure 3E and F). We then explain in more detail how the experimental perturbations changed the growth-progression parameters (beginning with “In terms of fit parameters…”). In particular, rapamycin slows the growth rate but also increases the strength of inheritance of the progression process, consistent with a picture where prolonged cycles also make e.g. dilution of cell-cycle inhibitors towards the end of the cycle more pronounced, as detailed in our response to the reviewer’s point 1) above. Thus, while growth-limited cycles are more frequent in presence of rapamycin, growth inhibition appears to also increase the inheritance of cell cycle factors, and together these effects produce the strong correlations shown in Figure 4D, including a mother-daughter correlation that is in fact higher than in MYCN inhibited conditions. We have added a statement to this effect in subsection “Cousin correlations reflect active cell-size checkpoint”.

Reviewer #2:[…]Main points:1) While the modelling results match the data within the estimated error bounds, the correlation patterns seem to be less pronounced. For example, the increased correlation to first cousins compared to aunts in Figure 3C (and Figure 3—figure supplement 1D) is hardly reproduced in the model. Similarly, the correlations to the removed side-branch seem to be about equal for all generations in the model, while the data indicates increased intra-generational correlations. It would be helpful if the authors comment on and explain such deviations or limitations of the model.

The reviewer points out potential differences in detail between correlation structures in the experimental data and in the model. We agree with this point. Both BAR model and growth progression models do not show an increase of correlations from aunts to cousins, although they do produce a moderate increase of correlation descending down the second side branch. As the reviewer also points out, the rather substantial measurement error bounds currently preclude us from going further in this direction. Even higher-precision data, or combination with other datasets (e.g., on detailed mechanisms of size regulation) may be necessary for further refining our models (for instance, resolving cell-cycle phases) in the future. We now indicate this in the Discussion.

2) As a main assumption of the authors is that MYCN over-expression disrupts the circadian clock which has been previously linked to the inheritance patterns of cell cycle length, they should validate this in the cell line used. This could be done for example by quantifying BMAL1 expression in the presence/absence of doxycycline (see Figure 4—figure supplement 1A).

The reviewer asks for additional evidence that the circadian clock is downregulated by high MYCN levels. In response, we compared the RNA levels of circadian clock genes in TET21N cells with high MYCN expression to their counterparts with downregulated MYCN (based on RNA-seq data obtained previously by us, Ryl et al., 2017). This analysis shows significant downregulation of four out of nine clock genes studied (*PER1, PER3, CRY2, NR1D1* downregulated); the other five genes (*ARNTL, PER2, CRY1, CLOCK, NR1D2*) are not significantly changed. At least two of those latter genes appeared moderately downregulated (*PER2, CLOCK*, but this was not significant) while none appeared upregulated.

As MYC/MYCN is known to induce moderate changes in transcript levels to a large number of genes, we asked whether the entire gene module of the circadian clock was coherently regulated. Applying a standard gene-module approach (Robinson and Smyth, 2008), we compared the transcript level of the circadian clock module formed by the nine clock genes to the expression of randomly drawn modules with similar dispersion of transcript levels as the clock module. In this comparison, the clock module was strongly (and significantly) downregulated. We have added these new data as Figure 1—figure supplement 4 and refer to this figure in the main text.

In a line of work separate from this manuscript, we engineered a stable dual-reporter TET21N cell line for imaging cell cycle (using the Cdt1 degron from the FUCCI markers) and circadian clock (fluorescent protein expression driven by *NR1D1*). In the first set of experiments, we found 1:1 phase locking with periods around 18 hours. As an example, Author response image 2 shows fluorescence traces of three cells. While analysis of more cell traces will be needed, this behavior is very similar to the 1:1 phase locking of cell cycle and circadian clock reported previously for mammalian cell lines (Bieler et al., 2014; Feillet et al., 2014), except that the clock in TET21N cells appears to be much noisier. The observed behavior is different from circadian gating or modulation of the cell cycle in cyanobacteria, which involves the circadian clock as an external oscillatory forcing with 24 hour period.

**Author response image 2. respfig2:** Sample traces of three TET21N cells with fluorescent reporters for the circadian clock (black lines) and the cell cycle (red lines). Dual reporter TET21N cells were grown and imaged as described in the manuscript.

Reviewer #3:Kuchen et al. have developed an interesting and general model that can explain the high degree of intra-generational correlation in cell cycle times that have been observed in multiple systems. First, using Bayesian inference on their own time-lapse data, they find that a model of two unidirectionally coupled heritable components that affect cell-cycle time best explains their data. They go on to propose that these components can be cell size and cell-cycle speed, provided that growth and division are coupled via a minimum size checkpoint. They then verify this model through perturbation experiments. Although the paper appears to be of general interest to eLife readers, there are several points that require clarification before publication.1) Representation of data. The authors present 3 repeats of the time-lapse microscopy for neuroblastoma TET21N cell lineages. (Figure 1E). For the Bayesian analysis (Figure 2) they only show the different models plotted against Repeat 1 in the main and supplemental figures (which appears to be the bottom dataset of Figure 1E – it would be helpful if the different repeats were labelled in this figure). However, there appears to be significant differences between the 3 repeats shown in Figure 1E. Although it is difficult to see (in part because the horizontal line across each axis is at a different value of the Cross correlation coefficient – how was this value selected?) it appears that in repeats 2 and 3 the level of correlation in distant side branches is much lower (around 0). This appears particularly worrying given the results of the Bayesian analysis for Model VII (Figure 2—figure supplement 1). Although this model results in no correlation past first cousins, it gives the best fit to data for repeat 2 and 3 (better in fact than model 5, Figure 2—figure supplement 1), and only fails to reproduce replicate 1. This suggests that for 2 out of 3 datasets a model that assumes no correlations past first cousins is being selected as the best model from the Bayesian analysis. The authors should address this issue, and show plots of the Bayesian analysis against all 3 repeats.

The reviewer begins by pointing out that experimental data are not optimally presented in the manuscript. In particular, there was an error in Figure 1E, where two of the x-axes were inadvertently drawn at the wrong y position. We apologize for this drawing error and have corrected it. We have also added replicate labels in this figure panel.

The reviewer then continues to point out that the three control repeats differ with respect to the extent of long-ranging intra-generational correlations and, indeed, the selection of BAR models yields different results for repeat 1 versus repeats 2 and 3. We agree that these are important points which were not fully discussed in the original manuscript.

We understood that this discussion arises largely due to the fact that the results on BAR model selection are split between main figure (Figure 2) and supplement (Figure 2—figure supplement 1). To rectify this problem, we have revised these figures as follows:

1) In the revised Figure 2, the complete selection of BAR models is shown. In the accompanying revision of the main text, we now discuss in more detail that the extent of long-ranging correlation differs between the control replicates and that this fact impacts model selection. Specifically, long-ranging correlations favor Model V over Model VII. To illustrate this, we show BAR model fits (including Model VII) versus repeat 1 in the main figure.

2) To further emphasize this point we have now added the fits of the different BAR models also for experimental repeats 2 and 3, as also requested by the reviewer. To not overload the main figure, these are now shown in Figure 2—figure supplement 1.

The key conclusion of our model selection is that the existence of long-range correlations selects BAR models that with two latent variables that show both self-inheritance and cross-inheritance, as exemplified by Model V. We exclude Model VII because it cannot account for the full data.

This conclusion is then corroborated subsequently in the manuscript by the rapamycin experiments with TET21N cells (Figure 4—figure supplement 1H) and by the ESC data (Figure 5—figure supplement 1D). where, again, Model VII is rejected due to long-range correlations. Note that in these special cases, Model IV (which is a more complex “version” of Model V) would be acceptable. Finally, the MYCN inhibited experiments are compatible and even prefer Model VII, due to the absence of long-range correlations (Figure 4—figure supplement 1G). Overall, only Model V can account acceptably for the full set of perturbed data, making it the only candidate for a model that describes a general mechanism. This selection results already when taking only the set of MYCN-high experiments rep1-3 into account.

We have corrected the mis-labeled caption of Figure 4—figure supplement 1. In the Discussion section we now added a sentence stating more precisely why the BAR Model V is favored overall, summarizing the above arguments.

In the rest of the figures, which dataset chosen is not consistent. Although they do display a fit of their growth/progression model to repeat 1 and 2 in the Figure 3—figure supplement 1, they now chose repeat 3 to display in Figure 3 – which appears to be the best fit to the model. In Figure 4, panel C shows cell-cycle lengths for repeat 3, but it is not clear which repeat(s) is/are used in the calculation of the correlation ratios. It would be helpful if the authors were consistent with their data selection in the main figures and supplements throughout the paper.

The reviewer points out that, in Figures 3 and 4, fits of the growth-progression model are shown for different experimental replicates while the fits to the remaining repeats are shown in the supplements to these figures. We agree that this may appear confusing and have now chosen in a consistent manner to show fits to replicate rep1 in the main figures – Figure 2 for the BAR model, Figure 3 for the growth-progression model, and again as control in Figure 4C (perturbation experiments) – while fits to repeats 2 and 3 are shown in the respective figure supplements. This reordering of figure panels does not influence the conclusions drawn (see above). Also, for the guidance of the reader, we added an overview of all time-lapse experiments used in the paper, as the new Figure 1—figure supplement 2.

2) Experimental evidence for cell size/cell cycle checkpoint model. The authors relate latent variables x_1_ and χ2 to cell size and the cycling of cell cycle factors respectively. However, since these quantities are not dynamically tracked throughout cell growth, I have some questions about the authors' interpretation and model. First, the model assumes the existence of a size threshold. If size does not affect cell cycle progression (another assumption), that suggests small-born cells are under a sizer-type control, whereas large-born cells (progression limited) are under a timer-type control. Such a model seems difficult to square with ongoing work in other organisms, which typically find adder-type (many single-cell microbes but also a number of mammalian cell lines, cf. Cadart et al., 2018) or sizer-type (e.g. fission yeast) control. Could the authors comment on how their model is biologically justified and how does it relate to alternative models of cell growth? Timer control is usually not considered because, on its own, it does not generate stable cell size.

The reviewer raises the important question of how our growth-progression model relates to recent experimental work on size control mechanisms in mammalian cells and fission yeast (a similar point was also raised by reviewer #1, Point 3).

The reviewer correctly points out that our growth-progression model combines “timer” cycles, when cycle duration is controlled by cell-cycle progression, and “sizer” cycles, when cycle duration is controlled by growth. Overall, the mix leads to a pattern of mitotic size versus birth size that lies between a pure timer and a pure adder (see Author response image 1). Indeed, Cadart et al. show that different mammalian cells fall within a continuum from timer to adder to sizer when quantifying cell size gain during the cycle. Hence, our specific model lies within this continuum, between timer and adder. It is important here to point out that timer, adder and sizer are phenomenological models that do not rely on actual molecular mechanisms of size control. It is conceivable that a specific molecular mechanism may generate something different from any such pure model. The fact that our growth-progression model (a phenomenological model that per se does not focus on size control) creates a pattern of cell size gain intermediate between pure timer and pure sizer is interesting in this respect. However, as the mechanistic basis of the fundamental findings by Cadart et al. is not fully understood, we feel it is beyond the scope of the current manuscript to address these questions. We indicate now in closing the Discussion that this may be a fruitful direction for future work.

Indeed, in the appendix the authors note that, in the progression-limited regime, their model generates cells that are too large, so they implemented logistic, rather than exponential growth. In my understanding, this effectively implements a sizer again. What evidence do the authors have for these assumptions?On the other hand, in subsection “Effects ofmolecular perturbations on cell-cycle correlations correctly predicted by the model” the authors claim that the growth of MYCN-inhibited cells is not affected. How did the authors verify this? If growth is logistic, as in the model, then growth rate is clearly affected beyond a certain size. Naively, I would also expect slower growth rates if growth is exponential and cell size is homeostatic.

The assumption of a minimum-size threshold in the growth progression model is clearly a parsimonious assumption intended to keep the number of model parameters to a minimum. There is now conclusive work showing that cells do sense their size and can adjust their growth rate as a result (e.g. Ginzburg et al., 2018). Indeed, we used such a mechanism – implemented as a logistic dependence of growth rate on size – to avoid size divergence when the fraction of growth-limited cycles becomes very small and progression-limited cycles dominate (as in the case of *MYCN* knockdown). Thus the critical question arises whether this type of sizer regulation would work in general in the growth-progression model. We have now tested this and found that incorporation of a logistic growth limitation does not significantly alter our results on cell-cycle correlations in lineage trees when cell cycles are faster, as in TET21N control conditions. This shows that our conclusion remain robust when feedback control of growth rate by size is incorporated.

Importantly, growth rate regulation kicks in at high values of the size variable that are very rarely reached when cells proliferate more rapidly. Hence some form of growth limitation will surely exist, but it is largely inconsequential for the more rapid cell cycles. Hence leaving out growth limitation for faster growing cells is a valid approximation.

These new results are shown in the new Figure 4—figure supplement 2 and commented on in the Results as well as in the last paragraph of the Discussion.

Additionally, the effects on cell size due to the addition of rapamycin appears weak (Figure 4B) – Is it possible to quantify this difference, with an estimation of errors?

Precise cell size (mass or volume) measurement is technically hard (see e.g., Tzur et al. 2009; Ginzberg et al., 2018; Cadart et al., 2018) and is not in the focus of the current paper. We used, as a robust proxy, the forward scatter in the flow cytometer. This does not allow quantification of size changes. We fully agree with the reviewer that the size decrease upon rapamycin treatment is small (we used doses of rapamycin that had a growth-inhibitory effect but did not stall cell cycling altogether). In response to the point by the reviewer, we now show two biological replicates side-by-side in amended Figure 4B for each condition (control, rapamycin, MYCN inhibition). These data show that the small size decrease upon rapamycin treatment is completely reproducible.